# 1 Measurement Report: Optical properties of supermicron

# 2 aerosol particles in a boreal environment

- 3 Sujai Banerji<sup>1</sup>, Krista Luoma<sup>2</sup>, Ilona Ylivinkka<sup>1</sup>, Lauri Ahonen<sup>1</sup>, Veli-Matti Kerminen<sup>1</sup>, and
- 4 Tuukka Petäjä<sup>1</sup>
- <sup>1</sup>Institute for Atmospheric and Earth System Research (INAR)/Physics, Faculty of Science, University of
- Helsinki, Helsinki, Finland
- <sup>2</sup>Finnish Meteorological Institute, Helsinki, Finland
- Correspondence to: Sujai Banerji (sujai.banerji@helsinki.fi)

# 9 Abstract

- Supermicron aerosol particles (PM<sub>1-10</sub>; here defined as 1  $\mu$ m < aerodynamic diameter  $\leq$  10  $\mu$ m) play a crucial role
- in aerosol-climate interactions by influencing light scattering and absorption. However, their long-term trends and
- episodic significance in boreal environments remain insufficiently understood. This study examines
- measurements of optical properties and mass of  $PM_{1-10}$  over a 12-year period at the SMEAR II station in Hyytiälä,
- Finland, focusing on their variability and key drivers. By assessing long-term trends, seasonality, and episodic
- variability, the study provides new insights into the role of these particles in aerosol-climate interactions. Episodic
- events, such as pollen outbreaks and dust transport, are identified as major contributors to PM<sub>1-10</sub> variability and
- their role in atmospheric processes. In addition, cascade impactor filters were used to quantify super-PM<sub>10</sub>
- particles  $(D_p > 10 \,\mu\text{m})$ , which are not detected by optical instruments, addressing key detection limitations. The
- 19 findings reveal significant long-term trends and pronounced seasonality in PM<sub>1-10</sub> mass and optical properties,
- 20 emphasizing their importance in boreal environments and their episodic relevance in coarse-mode aerosol
- 21 characterization.

22

#### 1. Introduction

- Aerosols are integral to atmospheric processes, influencing climate, air quality, and radiative forcing. Among
- 24 them, coarse-mode aerosol particles, which are typically defined as particles with diameters  $> 1 \mu m$  play a
- significant role in light scattering and absorption, directly impacting radiative forcing. Their size and optical
- properties make them dominant contributors to aerosol optical depth (AOD), particularly at longer wavelengths.
- Coarse-mode aerosol particles, such as biological aerosols found in the boreal environment, also contribute to
- cloud microphysics by serving as ice-nucleating particles (Brasseur et al., 2022). Despite their importance, the
- optical properties and particulate matter mass (PM mass) of coarse-mode aerosol particles remain understudied
- (Cappa et al., 2016).
- Boreal forests, covering approximately 15% of the Earth's terrestrial surface, represent a unique natural laboratory
- for studying aerosol-climate interactions in biogenically dominated environments. These ecosystems emit large
- quantities of biogenic volatile organic compounds (BVOCs) Guenther et al. (2006), which drive the formation of

secondary organic aerosols (SOA), significantly influencing aerosol size distribution and, therefore, light scattering and absorption processes (Petäjä et al., 2022; Tunved et al., 2006). Additionally, episodic events such as pollen outbreaks and long-range transport of mineral dust contribute to aerosol variability in boreal regions (Manninen et al., 2014).

Coarse-mode aerosol particles in boreal environments are emitted predominantly through primary processes rather than formed secondarily, arising from mineral dust, pollen, fungal spores, plant debris, sea salt, and episodic sources such as wildfires and small-scale wood combustion (Zieger et al., 2015; Yli-Panula et al., 2009; Varga et al., 2023; Andreae and Merlet, 2001; Reid et al., 2005). Wildfires are predominantly a summer phenomenon, whereas small-scale wood combustion peaks in winter and likely explains the higher winter concentrations. Dust commonly reaches Finland via long-range transport, whereas pollen and fungal spores are locally or regionally emitted and highly seasonal (Varga et al., 2023; Yli-Panula et al., 2009). Marine sea-salt intrusions occasionally affect inland boreal forests during strong winds or frontal passages (Zieger et al., 2015; Tunved et al., 2006).

Coarse-mode aerosol particle sizes span about 1  $\mu$ m to > 10  $\mu$ m, so many pollen grains and fungal spores exceed the PM<sub>10</sub> impactor cut-off and are underrepresented in PM<sub>10</sub> measurements (Després et al., 2012; Yli-Panula et al., 2009). This selectivity can bias coarse-mode analyses during episodic biological events, unless sampling and interpretation account for partial capture or exclusion. Determining whether such particles are sampled is crucial for quantifying impacts on aerosol optical properties, radiative forcing, and aerosol-cloud interactions (Zieger et al., 2015; Tunved et al., 2006). Wildfire smoke produces fine-mode particles but can include coarse fractions from smouldering or resuspension (Andreae and Merlet, 2001; Reid et al., 2005).

The natural components of coarse-mode aerosol particles contribute to spatial and temporal heterogeneity, influencing key microphysical processes, such as the activation of ice-nucleating particles (INPs) and playing a vital role in atmospheric processes like cloud formation, nutrient cycling and radiative interactions (Brasseur et al., 2024; Després et al., 2012; Mahowald et al., 2014; Schneider et al., 2021). Despite their recognized importance, the interactions between these natural components and atmospheric dynamics in boreal environments remain poorly characterized, particularly in the context of episodic events like pollen releases, dust transport and potential contributions from biomass burning.

The coarse-mode size range also presents challenges for optical instruments, with upper size cut-offs typically set to  $10~\mu m$ , potentially resulting in a significant underestimation of aerosol mass (Zieger et al., 2015). This undetected mass fraction is crucial for achieving optical closure and improving vertical column density estimates in boreal environments. Furthermore, the detection of coarse-mode aerosol particles depends on their size—particles  $> 10~\mu m$ , Smaller coarse-mode particles like fungal spores and dust are more likely to be captured, whereas larger ones, such as pollen, may escape detection.

The knowledge gaps regarding coarse-mode aerosol particles hinder our ability to fully understand their optical and mass properties and their contributions to regional and global atmospheric processes. To capture both long-term trends and episodic variability, we analyze the optical properties and PM mass of PM<sub>1-10</sub> aerosol particles at the SMEAR II station in Hyytiälä, Finland. We also quantify the PM mass of super-PM<sub>10</sub> particles ( $D_p > 10 \mu m$ )

- via gravimetric analysis of cascade impactor filters to contextualize their sources, variability and impacts.
- Specifically, this study aims to:
- (a) Investigate the measurements of optical properties and mass of  $PM_{1-10}$  aerosol particles at the SMEAR II
- station to analyze long-term trends of coarse-mode aerosol particles and their variability in boreal environments.
- (b) Quantify the PM mass fraction of coarse-mode aerosol particles beyond the detection limits of optical
- instruments, constrained by the 10 µm size cut-off (super-PM<sub>10</sub>), with implications for optical closure and vertical
- column density estimates.
- (c) Assess the statistical significance of episodic events, such as pollen outbreaks and dust transport, on the optical
- properties of  $PM_{1-10}$  aerosol particles.
- (d) Examine the PM mass of the super-PM<sub>10</sub> aerosol particles to fully capture the contributions of larger particles
- and their episodical significance in events such as pollen outbreaks or dust transport.

## 2. Measurement and methods

#### 82 2.1 SMEAR II

- The Station for Measuring Ecosystem-Atmosphere Relations (SMEAR II) is located in Hyytiälä, southern Finland
- (61°51'N, 24°17'E; 181 m.a.s.l.). It is located in a boreal forest and is classified as a background site where no
- major local sources of aerosol particles from anthropogenic activities are observed (Hari & Kulmala, 2005).
- However, there are some sources of pollution in the region, such as the city of Tampere (60 km in the southwest
- direction with a population of 241,000 in 2021) and the activity of the buildings in the station (Boy, 2004; Kulmala
- et al., 2001). Therefore, SMEAR II reflects typical boreal forest conditions (Hari et al., 2013). It has instruments
- to measure interactions between the forest ecosystem and the atmosphere and the station is part of ACTRIS (Laj
- et al., 2024). The station has various aerosol instruments, including the four instruments used in this study: two
- Magee Scientific aethalometers (models AE31 and AE33) to obtain the light absorption coefficient at seven
- wavelengths, a TSI integrating nephelometer (model 3563) to calculate the light scattering coefficient at three
- wavelengths and a Dekati cascade gravimetric impactor to measure the PM mass of different size fractions (i.e.
- $\leq$  PM<sub>1</sub>, between PM<sub>1</sub> and PM<sub>2.5</sub>, between PM<sub>2.5</sub> and PM<sub>10</sub>,  $\leq$  PM<sub>10</sub>, > PM<sub>10</sub>). The instruments are described in
- more detail in the following sections. All these instruments are in a 'Hitumökki,' i.e., the 'Aerosol Cottage.'

## 2.2 Measurement setup and instruments

- The measurement setup for the aerosol optical instruments included a pre-impactor designed to remove the aerosol
- particles with a  $D_p > 10 \,\mu\text{m}$  sampling PM<sub>10</sub> aerosol particles. The inlet is located at a height of 8 m above ground
- inside the forest canopy. Following the pre-impactor, the airflow sequentially passed through an inlet flow splitter.
- This configuration allowed us to sample aerosol particles with aerodynamic diameter <1 µm (hereafter PM<sub>1</sub>
- aerosol particles) or PM<sub>10</sub> every ten minutes. The switching inlet system has been described in detail by Luoma
- et al. (2021). Subsequently the size selected sample with a flow rate of 30 l min<sup>-1</sup> is split into three streams to
- optical instruments.

- One of the airstreams from the splitter goes into a Nafion dryer connected to an integrating nephelometer (TSI
- model 3563). With the Nafion dryer, the relative humidity of the sampled air is aimed to be kept ≤40%. The
- integrating nephelometer maintains a flow rate of 8.3 lmin<sup>-1</sup>.
- The second airstream from the splitter passes through a different Nafion dryer and then enters the Magee Scientific
- aethalometer. The AE31 operated at SMEAR II until December 2017 and was replaced by the newer model AE33
- in February 2018. The aethalometers maintain a flow rate of 5 lmin<sup>-1</sup>.
- The third airstream is directed into a Thermo Fischer multi-angle absorption photometer (MAAP; model 5012)
- through a Nafion dryer. The MAAP maintains the flow rate of 16.7 lmin<sup>-1</sup>.

#### 2.2.1 Aethalometer

- The aethalometer quantifies the aerosol absorption coefficient ( $\sigma_{abs}$ ) by measuring the reduction in light intensity
- as particles collect on a filter, facilitating continuous aerosol sampling (Zotter et al., 2017). The AE31 and AE33
- models compare photon counts from light transmitted through a particle-laden filter spot to a clean reference filter.
- Correction algorithms account for aerosol particle scattering and multiple scattering within the quartz fiber filter.
- As light-absorbing particles build up, the effective optical path length shortens, necessitating adjustments for the
- filter-loading effect (Collaud Coen et al., 2010; Weingartner et al., 2003). The multiple-scattering correction factor
- $(C_{ref})$  addresses the enhancement of light scattering within the filter matrix due to the filter material, while the
- filter loading correction factor (R(ATN)) accounts for the non-linear instrument response caused by particle
- accumulation on the filter (Liousse et al., 1993). Since then, several studies have refined these correction
- approaches (Weingartner et al., 2003; Virkkula et al., 2007; Collaud Coen et al., 2010; Drinovec et al., 2015; Yus-
- Díez et al., 2021; Luoma et al., 2021).
- The Magee Scientific AE31 Aethalometer was operated at SMEAR II from October 2010 to December 2017 for
- continuous measurements of  $\sigma_{abs}$  at seven discrete wavelengths between 370 and 950 nm. Due to the single-spot
- filter design of the AE31, post-processing corrections were applied to account for artifacts arising from both
- multiple scattering within the quartz fiber filter matrix and the filter-loading effect. In this study, a  $C_{ref}$  of 1.57
- was uniformly applied across all wavelengths, following the methodology presented by (Luoma et al., 2021). This
- value was derived using the correction algorithm developed by (Arnott et al., 2005), based on comparisons with
- absorption measurements from a MAAP at the same site. The R(ATN) was also applied to compensate for the
- reduction in the effective optical path length as particles accumulated on the filter. These corrections were essential
- for ensuring accurate  $\sigma_{abs}$  retrievals and for addressing known biases in AE31 measurements associated with the
- scattering properties of the filter substrate. The corrected data enable reliable characterization of aerosol light
- absorption and its spectral dependence in the boreal environment.
- From January 2018 to October 2022, the AE33 replaced the AE31 at SMEAR II. The AE33's dual-spot design
- corrects filter loading online (Drinovec et al., 2015), eliminating post-processing steps such as R(ATN). In the
- AE33, the  $C_{ref}$  is a fixed, user-defined setting determined *apriori*—primarily by the filter tape—and remains
- constant during operation (AE33 User Manual, V1.56, August 2018). For consistency, we used two fixed values:
- $C_{ref} = 1.57$  for 2010–17 (AE31; Luoma et al., 2019) and  $C_{ref} = 1.39$  for 2018–22 (AE33 with Magee M8060). Yus-
- Díez et al. (2021) showed that  $C_{ref}$  depends on both tape and aerosol optical properties; because their dataset did

not include a low-altitude boreal background site comparable to SMEAR II, we did not apply additional site-specific or SSA-based adjustments. The key difference between the two instruments lies in their handling of the filter-loading effect. The AE31 requires manual post-processing to address non-linear responses caused by particle accumulation, while the AE33 performs these corrections in real time, reducing the need for post-processing and enhancing data reliability. This distinction, combined with the AE33's improved algorithms, allows for more accurate and consistent measurements of aerosol properties in dynamic environments.

#### 2.2.2 Multi-angle absorption photometer

The MAAP measures light absorption coefficient and equivalent black carbon concentration at a 637 nm wavelength ( $\sigma_{abs,637}$  and eBC, respectively) with a one-minute resolution and maintains a 16.67 lmin <sup>-1</sup> airflow using an external pump. It collects aerosol particles on glass fiber filter tape (Saturno et al., 2017). When particle accumulation reaches a threshold, the tape advances to a new place to avoid saturation. A 637 nm light source measures transmitted photon counts at a 0° detection angle, while reflected counts are measured at 130° and 165° to assess hemispheric backscattering (Petzold et al., 2005; Petzold & Schönlinner, 2004). The aerosol layer and filter matrix are modeled as a two-layer system, using the adding method and radiation budget equations (Petzold et al., 2005; van de Hulst, 1980). These equations are solved iteratively using the *SSA* of the aerosol-loaded filter layer and layer optical depth until consistent values are reached (Petzold & Schönlinner, 2004). The *SSA* is then used to calculate the aerosol mass and black carbon mass per unit volume, leading to the determination of  $\sigma_{abs,637}$ .

## 2.2.3 Integrating nephelometer

The TSI Incorporated model 3563 integrating nephelometer measures the light scattering and backscattering coefficients ( $\sigma_{sca}$  and  $\sigma_{bsca}$ , respectively) of the aerosol particles in their airborne state (Anderson et al., 1996). The instrument comprises three primary elements: a measurement chamber, a light source, and a detector. A diffuser guarantees that the light source emits a wavefront resembling a Lambertian distribution into the chamber. The chamber has a detector at one end and a light trap at the other. The chamber's interior is covered with black flocked paper to absorb stray light. The reference chopper positioned in front of the detector consists of three distinct sections: a signal section responsible for integrating light scattering within the range of  $7^{\circ}$  to  $170^{\circ}$ , a dark portion used to measure background noise, and a calibration section utilized to ensure the stability of the light source. A correction to address the scattering in blind angles ( $<7^{\circ}$  and  $>170^{\circ}$ ) is applied according to Anderson and Ogren (1998).

A revolving backscatter shutter obstructs light within a scattering angle range of 7-90°, allowing for the measurement of hemispheric backscattering between 90° and 170°. The detector consists of three photomultiplier tubes and a lens that aligns dispersed light into parallel rays, dividing it into wavelengths of 450, 550, and 700 nm using dichroic and bandpass filters. The  $\sigma_{sca}$  and  $\sigma_{bsca}$  are determined by integrating the simplified scattering phase function. The data are corrected for gas molecule scattering by employing a HEPA filter. Regular span checks using CO<sub>2</sub> gas are conducted to account for instrument drift (Anderson et al., 1996).

## 2.2.4 Cascade impactor with subsequent gravimetric analysis

The sampling of ambient aerosol particles at SMEAR II has been conducted since the late 1990s using a Dekati PM<sub>10</sub> cascade impactor filter with an unheated inlet for total suspended particulates (Laakso et al., 2003; Petäjä et al., 2025). The inlet, vertically sampling from approximately 5 m above ground level, consists of a stainless-steel tube with a rain cover. The impactor separates particles into three size fractions with aerodynamic diameter cut points at 10 μm (PM<sub>10</sub>), 2.5 μm (PM<sub>2.5</sub>), and 1 μm (PM<sub>1</sub>) across its three stages. This separation is achieved with a consistent volumetric air flow rate of 30 l min<sup>-1</sup> (Berner & Luerzer, 1980).

Collection substrates for the first two stages include 25 mm polycarbonate membranes (Nuclepore 800 203) without perforations. In contrast, the final stage employs a 47 mm Teflon filter with a 2 µm pore size (R2P J047) from Pall Corporation. To minimize particle rebound from the collection surfaces, the membranes are coated with a thin layer of Apiezon L vacuum grease and diluted in toluene. After collection, particulate samples are weighed in gravimetric analysis.

Particulate samples are weighed to produce aerosol mass distribution in two-to-three-day averages. The mass distributions are calculated based on the differences in filter weights before and after sampling. Once weighed, the filters are stored in a freezer to preserve the samples for future chemical and physical analyses. This procedure ensures the reliable collection, quantification, and archiving of aerosol particle data for long-term studies. The PM mass data across different aerosol particle size ranges, i.e.  $PM_1$ ,  $PM_{10}$  and super- $PM_{10}$  was merged with the hourly resampled aerosol optical properties (AOPs).

## 2.3 Data processing of aerosol optical and PM mass-related properties

The aerosol optical data analyzed in this study span 4 October 2010 to 4 October 2022. Optical properties for the  $PM_{1-10}$  size fraction were obtained by subtracting  $PM_1$  measurements from the corresponding PM10 measurements of scattering and absorption. All optical time series were first resampled to hourly averages with a +30-minute offset, yielding timestamps centered at HH:30 (e.g., 00:30:00, 01:30:00, 02:30:00). This offset follows the EBAS convention maintained by NILU, in which hourly values are centered within their averaging intervals. The native temporal resolution of each instrument is listed in Table 1 and is preserved in the metadata; hourly resampling is used solely to align instruments with differing sampling cadences prior to subsequent aggregation.

To enable comparison with the impactor *PM mass* measurements, the optical data were averaged over the full duration of each impactor filter sampling interval, defined by the start—end timestamps of the sampling. For every interval, all optical points within the bounds were identified and averaged arithmetically, and the interval midpoint was used as the representative timestamp. The corresponding *PM mass* measurements were mapped onto the same intervals. Because filter samples typically span 2 to 3 days, these aggregated values are hereafter referred to as pseudo-daily mean values. This harmonization follows established practice for combining datasets with differing temporal resolutions and underpins the episodic and long-term analyses (Sheridan and Ogren, 1999; Collaud Coen et al., 2020); pollen events were classified from the 2–3-day filter samples, whereas dust events followed the day-specific scheme of Varga et al. (2023).

Additional processing ensured consistency across size cuts and instruments. All *PM mass* related-variables (*PM mass, MSC*<sub>550</sub>: mass scattering coeffcient at 550 nm and MAC<sub>520</sub>: mass absorption coefficient at 520 nm) were normalized by co-located dry *PM mass*, i.e. relative humidity (RH)  $\leq$  40%, for the same size fraction (PM<sub>10</sub>, PM<sub>1</sub> and PM<sub>1-10</sub>), with PM1–10 formed at the hourly step as PM<sub>10</sub> and PM<sub>1</sub>. prior to aggregation to the impactor filter intervals. Scattering coefficients were measured at low RH and corrected for angular truncation and non-ideal angular response (Anderson and Ogren, 1998; Müller et al., 2011). Light absorption coefficient at 520 nm ( $\sigma_{abs,520}$ ) was obtained from AE31 (2010–2017) and AE33 (2018–2022), with the AE33 dual-spot loading correction applied as provided (Drinovec et al., 2015). These harmonized and corrected optical timeseries, aligned to the filter sampling windows, provide a consistent basis for the subsequent seasonal and trend evaluations of both optical and mass-based aerosol properties (Sheridan and Ogren, 1999; Collaud Coen et al., 2020; Varga et al., 2023; Anderson and Ogren, 1998; Müller et al., 2011; Drinovec et al., 2015).

**Table 1.** Temporal resolutions and size cut-offs of the different aerosol optical instruments

| Instrument                                                             | Temporal resolution | Size cut-off                                                                                              |
|------------------------------------------------------------------------|---------------------|-----------------------------------------------------------------------------------------------------------|
| Aethalometer (AE33)                                                    | 2 minutes           | PM <sub>1</sub> , PM <sub>10</sub>                                                                        |
| Multi angle absorption photometer (MAAP; Thermo Scientific model 5012) | 1 minute            | PM <sub>1</sub> , PM <sub>10</sub>                                                                        |
| Integrating nephelometer (TSI Incorporated model 3563)                 | 1 minute            | PM <sub>1</sub> , PM <sub>10</sub>                                                                        |
| Dekati Gravimetric Cascade Impactor (GCI)                              | 2-3 days            | $ \leq PM_{1}, \leq PM_{1}\text{-}PM_{2.5}, \leq \\ PM_{2.5}\text{-}PM_{10}, \leq PM_{10}, > \\ PM_{10} $ |

In this study, we analyzed absorption data for both PM<sub>1</sub> and PM<sub>10</sub> aerosol particles. Specifically, we utilized the MAAP data to plot  $\sigma_{abs,637}$ , and the AE31/33 data to plot  $\sigma_{abs,660}$ . Subsequently, a scatterplot was created with the MAAP data to plot  $\sigma_{abs,637}$  on the x-axis and the AE31/33 data to plot  $\sigma_{abs,660}$  on the y-axis. This allowed us to determine the slope and y-intercept, which were then used to scale  $\sigma_{abs,660}$  data from the AE31/33 instruments to match  $\sigma_{abs,637}$  from the MAAP instrument. The same correction factors (slope = 2.33 and y-intercept = -0.16; Figure S9) derived from this analysis were extended to scale the  $\sigma_{abs}$  data from the AE31/33 instrument at the other six wavelengths of 370, 470, 520, 550, 880 and 950 nm to correct all the data from the AE31/33 instruments.

For calculating the trends in the different aerosol optical data, the Mann-Kendall regression was used to effectively handle outliers without assuming a normal distribution (Collaud Coen et al., 2020). This method also minimizes the risk of Type I errors, which can occur when a trend is incorrectly identified as 'statistically significant' due to an anomaly in the data (i.e., autocorrelation). Additionally, Sen-Theil's slope estimator has been used to quantify the slope of the trends, providing a reliable measure of the long-term changes in the aerosol optical properties, even if there are non-linear trends and seasonal fluctuations in the data (Collaud Coen et al., 2020).

Equation (1) was used to calculate the relative slope:

Relative slope (%yr<sup>-1</sup>) = 
$$\left(\frac{\text{Sen-Theil's slope}}{\text{Median of aerosol optical data}} \times 365.25 \text{yr}^{-1} \times 100\%\right)$$
, (1)

- Two key parameters used to characterize the wavelength dependence of the AOPs are the Absorption Ångström
- Exponent (AAE) and the Scattering Ångström Exponent (SAE). These exponents provide insights into aerosol
- composition and particle size distributions, offering indirect information about the types of aerosols present. While
- they are not directly used to estimate climate effects, they are important for understanding the physical and
- chemical properties of aerosols, which influence their behavior and interactions with radiation.

# 243 2.3.1 Absorption Ångström exponent (AAE)

- The AAE represents the wavelength dependence of aerosol light absorption and provides insights into the chemical
- characteristics of aerosol particles, such as brown carbon (BrC) or presence of coatings on a BC core, which have
- important implications for radiative forcing (Cazorla et al., 2013). The AAE was calculated by determining the
- slope of the ordinary least squares (OLS) linear fit of the natural logarithmic values of the  $\sigma_{abs}$  as a function of
- wavelength:

$$AAE = -\frac{\Delta \ln(\sigma_{\text{abs}, \lambda})}{\Delta \ln(\lambda)},$$
 (2)

- where  $\sigma_{abs}$  represents the light absorption coefficient at wavelength  $\lambda$ . For AAE, absorption measurements are
- taken at the following 7 wavelengths: 370, 470, 520, 590, 660, 880 and 950 nm.

# 252 2.3.2 Scattering Ångström exponent (SAE)

- The SAE quantifies the wavelength dependence of aerosol light scattering and is closely associated with aerosol
- particle size distribution. Higher SAE values generally indicate a dominance of smaller, fine-mode particles, which
- exhibit a stronger wavelength dependence in scattering behavior. Conversely, lower SAE values are associated
- with larger, coarse-mode particles that scatter light more uniformly across wavelengths. This relationship makes
- the SAE a valuable metric for assessing aerosol size distributions, providing insight into the relative abundance of
- fine and coarse particles and their implications for atmospheric radiative properties and climate-forcing (Schuster
- et al., 2006).
- In this study, the SAE was calculated similarly to the AAE by applying an Ordinary Least Squares (OLS) linear
- regression to the natural logarithmic values of the light scattering coefficients at 450, 550, and 700 nm, regressed
- against the natural logarithmic values of these wavelengths, as shown in Equation 3:

$$SAE = -\frac{\Delta \ln(\sigma_{sca}, \lambda)}{\Delta \ln(\lambda)},$$
(3)

# 264 2.3.3 Single scattering albedo (SSA)

- The SSA is a key aerosol optical property that is defined as the ratio of the  $\sigma_{sca}$  to the total extinction coefficient
- (the sum of  $\sigma_{sca}$  and  $\sigma_{abs}$ ). The SSA is instrumental in determining whether aerosols exert a net cooling or warming

effect on the atmosphere: higher *SSA* values generally indicate that aerosols are primarily scattering, contributing to a cooling effect, while lower *SSA* values suggest a more significant role in absorption, which leads to warming (Bond et al., 2013; Luoma et al., 2019; Tian et al., 2023). *SSA* can be calculated at any wavelength; in this study, *SSA* was calculated explicitly at 550 nm (hereon *SSA*<sub>550</sub>).

# 2.3.4 Mass absorption coefficient (MAC)

The MAC represents the efficiency with which aerosol particles absorb light per unit mass (not to be confused with MAC of BC, which is used in conversion of  $\sigma_{abs}$  to BC mass). The MAC is sensitive to changes in aerosol chemical composition and is a valuable indicator of shifts in absorbing components, such as BC and certain light absorbing organic compounds (i.e., BrC). These changes in the MAC often reflect variations in aerosol sources or chemical aging processes (Andreae & Gelencsér, 2006; Bond & Bergstrom, 2006). Long-term studies on MAC can reveal trends in aerosol composition, especially in regions like the Arctic and boreal environments, where variations in pollution sources and climate-driven changes are prevalent.

## 2.3.5 Mass scattering coefficient (MSC)

The *MSC* quantifies the efficiency with which aerosol particles scatter light per unit mass. In contrast to the *MAC*, the *MSC* is primarily influenced by the physicochemical characteristics of aerosols—especially particle size, shape, and composition—as these properties impact light-scattering efficiency per unit mass (Bates et al., 2005; Seinfeld & Pandis, 2016). Higher *MSC* values are typically associated with a more significant proportion of scattering particles, such as sulfate, nitrate, and other non-absorbing components, which contribute to a cooling effect in the planetary boundary layer (PBL), partially counteracting the warming effect of absorbing aerosols (Pandolfi et al., 2014).

# 3. Results and discussion

## 3.1 General characteristics of PM<sub>1-10</sub> aerosol particles

Long-term observations at SMEAR II indicate that  $PM_{1-10}$  primarily contributes to aerosol scattering, with minimal absorption due to its composition, which is predominantly influenced by aerosol particles of biogenic origin (e.g., pollen, fungal spores) and mineral dust (Luoma et al., 2019; Zieger et al., 2015). The SSA remains consistently high (>0.90), while the MSC is lower than that of  $PM_1$ , aligning with Mie theory predictions for larger particles (Pandolfi et al., 2018; Titos et al., 2021). A summary of these descriptive statistics for PM1-10 aerosol particles is provided in Table 2. Seasonal variations reveal increased  $PM_{1-10}$  scattering during spring and early summer due to enhanced biogenic activity, particularly from pollen and fungal spores (Heikkinen et al., 2020; Yli-Panula et al., 2009), whereas winter conditions favor fine-mode aerosols, reducing the relative contribution of  $PM_{1-10}$  (Luoma et al., 2021).

Intermittent mineral dust intrusions, predominantly from the Aral-Caspian region, contribute to episodic increases in atmospheric dust concentrations over Finland (Varga et al., 2023). While these events introduce coarse-mode aerosols, their influence on the absorption properties of PM<sub>1-10</sub>, particularly in terms of MAC variability, remains

poorly characterized due to limited long-term observations and uncertainties in aerosol source contributions. Lihavainen et al. (2015) analyzed long-term aerosol optical properties at the Pallas Global Atmospheric Watch station, focusing on  $PM_{10}$  rather than specifically  $PM_{1-10}$ , but their findings on seasonal variations in scattering and absorption provide useful context for interpreting boreal aerosol trends.

Hygroscopic growth measurements suggest that  $PM_{1-10}$  is less effective as cloud condensation nuclei (CCN) compared to fine-mode aerosols, influencing its atmospheric lifetime and radiative effects (McFiggans et al., 2006). While Lihavainen et al. (2015) examined aerosol hygroscopic properties in northern Finland, their study primarily covered  $PM_{10}$  rather than isolating the  $PM_{1-10}$  fraction. The contribution of  $PM_{1-10}$  to total  $PM_{10}$  scattering at SMEAR II remains substantial, though its representation in optical measurements is uncertain due to limitations in nephelometry and aethalometry, which tend to underestimate coarse-mode aerosol properties (Brasseur et al., 2024; Zieger et al., 2015).

**Table 2.** Descriptive statistics of the aerosol optical properties for all the valid data of the PM<sub>1-10</sub> particles.

|                        |                              | Wavelength (nm) | PM <sub>1-10</sub> aerosol particles |      |      |      |       |  |
|------------------------|------------------------------|-----------------|--------------------------------------|------|------|------|-------|--|
| Extensive variables    |                              |                 | 25%ile Mean Median 75%ile Std.       |      |      |      |       |  |
|                        | $\sigma_{abs}(Mm^{-1})$      | 520             | 0.07                                 | 0.17 | 0.12 | 0.21 | 0.18  |  |
|                        | $\sigma_{sca} (Mm^{-1})$     | 550             | 1.49                                 | 3.15 | 2.39 | 3.92 | 2.78  |  |
|                        | PM mass (µgm <sup>-3</sup> ) |                 | 0.90                                 | 2.20 | 1.50 | 2.45 | 3.22  |  |
| Intensive<br>variables | AAE<br>(dimensionless)       | 370-950         | 0.54                                 | 0.73 | 0.73 | 0.91 | 0.45  |  |
|                        | SAE<br>(dimensionless)       | 450-700         | 0.13                                 | 0.38 | 0.36 | 0.69 | 0.60  |  |
|                        | SSA<br>(dimensionless)       | 550             | 0.93                                 | 0.93 | 0.95 | 0.96 | 0.37  |  |
|                        | $MAC(m^2g^{-1})$             | 520             | 1.03                                 | 2.03 | 1.58 | 2.52 | 10.79 |  |
|                        | $MSC(m^2g^{-1})$             | 550             | 0.04                                 | 0.12 | 0.08 | 0.13 | 0.91  |  |

## 3.2 Long-term trends of extensive properties

Extensive properties refer to the properties of aerosol particles that depend on the amount of the aerosol particles. From the studied extensive properties,  $\sigma_{sca,550}$ ,  $\sigma_{abs,520}$  and mass for PM<sub>1-10</sub> aerosol particles show a clear long-term negative trend from October 2010 to October 2022, reflecting a decline in optical and mass properties (Figure. 1). To compare the relative differences with PM<sub>1-10</sub>, trends for PM<sub>1</sub> and PM<sub>10</sub> size fractions are also studied, and all the trends are presented in Table 3.

This decrease aligns with earlier findings at SMEAR II, as reported by Luoma et al. (2019), where reductions in extensive properties were attributed to declines in particle number concentration and volume concentrations, particularly impacting larger accumulation mode and coarse-mode aerosol particles with peaks at approximately 700 nm and  $5 \mu m$ , respectively.

Pollen and dust events (green and red stars; Sect. 2.3) introduce additional short-term variability in the observed optical and mass properties without altering the sign or statistical significance of the long-term trends (Table 3). Pollen events are typically seasonal, occurring during spring and summer, and contribute significantly to particle mass and scattering in the  $PM_{1-10}$  size range. Dust events, which are more intermittent and often result from long-range transport, enhance scattering and mass concentrations in the coarse mode, including  $PM_{1-10}$  aerosol particles. While these events introduce short-term variability, they do not alter the overall long-term declining trends. The pseudo-daily mean values used in this analysis help to better highlight such events by providing a finer temporal resolution, enabling the identification of short-term peaks in optical and mass properties alongside the broader trends.

**Table 3.** Long-term trends and statistical significance of  $PM_{1-10}$  aerosol optical and  $PM_{1-10}$  mass properties.

| S. No. | Variable                           | Slope                                                                                             | Relative trend                    | p-value                 | Statistical significance |
|--------|------------------------------------|---------------------------------------------------------------------------------------------------|-----------------------------------|-------------------------|--------------------------|
| (a)    | $\sigma_{sca, 550}(Mm^{-1})$       | $-0.05 \pm 0.04 \text{ Mm}^{-1} \text{yr}^{-1}$                                                   | -1.93 ± 1.74 %yr <sup>-1</sup>    | 0.16                    | No                       |
| (b)    | $\sigma_{abs, 520}(Mm^{-1})$       | $-9.83 \times 10^{-3} \pm 2.28 \times 10^{-3}$<br>$^{3}$ Mm <sup>-1</sup> yr <sup>-1</sup>        | -8.02 ± 1.86 %yr <sup>-1</sup>    | 0.052                   | No                       |
| (c)    | SAE<br>(dimensionless)             | $-0.03 \pm 0.02 \text{ yr}^{-1}$                                                                  | $-7.30 \pm 4.16 \text{ %yr}^{-1}$ | 0.08                    | No                       |
| (d)    | AAE (dimensionless)                | $0.03 \pm 8.63 \times 10^{-3} \mathrm{yr^{-1}}$                                                   | $3.54 \pm 1.04 \text{ %yr}^{-1}$  | 1.78 x 10 <sup>-3</sup> | Yes                      |
| (e)    | PM mass (µgm <sup>-3</sup> )       | $-0.04 \pm 0.01  \mu gm^{-3} yr^{-1}$                                                             | -2.61 ± 1.00 %yr <sup>-1</sup>    | 0.06                    | No                       |
| (f)    | SSA <sub>550</sub> (dimensionless) | 1.94 x 10 <sup>-3</sup> ± 7.75 x 10 <sup>-4</sup> yr <sup>-1</sup>                                | $0.20 \pm 0.08 \text{ %yr}^{-1}$  | 0.10                    | No                       |
| (g)    | $MSC_{550} (m^2 g^{-1})$           | $-0.02 \pm 0.01 \text{ m}^2\text{g}^{-1}\text{yr}^{-1}$                                           | -1.13 ± 0.94 %yr <sup>-1</sup>    | 0.50                    | No                       |
| (h)    | $MAC_{520} (m^2 g^{-1})$           | $-2.77 \times 10^{-3} \pm 9.04 \times 10^{-3}$<br>$^{4} \text{ m}^{2}\text{g}^{-1}\text{yr}^{-1}$ | -3.97 ± 1.30 %yr <sup>-1</sup>    | 0.02                    | Yes                      |

Figure 1. Time series of (a)  $\sigma_{sca,550}$ , (b)  $\sigma_{abs,520}$  and (c) *PM mass* for the PM<sub>1-10</sub> size aerosol particles from October 2010 to October 2022. The blue shaded area is the interquartile range (25th–75th percentile) of monthly values; the blue line is the monthly median. The red line shows the Theil-Sen trend (solid if p  $\leq$  0.05; dashed otherwise). Months with <75% data coverage are left blank.

## 3.2.1 Light scattering coefficient at 550 nm ( $\sigma_{sca,550}$ )

Scattering due to PM<sub>1</sub> aerosol particles at SMEAR II shows a long-term decrease (slope:  $-0.29 \pm 0.15$  Mm<sup>-1</sup>yr<sup>-1</sup>; relative trend:  $-4.80 \pm 2.43$  %yr<sup>-1</sup>; Figure S2(a); Table S4(a)). Submicron particles dominate aerosol light scattering at the site (Virkkula et al., 2011), suggesting that  $\sigma_{sca,550}$  is influenced primarily by fine-mode aerosol loading. The observed negative trend is consistent with reductions in anthropogenic sulfur dioxide (SO<sub>2</sub>) emissions, which contribute to secondary sulfate formation, a major component of PM<sub>1</sub> aerosol mass and associated optical properties (Smith et al., 2011). Regionally, similar decreases in aerosol scattering have been reported at multiple European background stations (Pandolfi et al., 2018), further supporting the possibility of a widespread decline in fine-mode aerosol scattering. SMEAR II is also subject to seasonal biogenic emissions, particularly monoterpenes that contribute to SOA formation (Hakola et al., 2003; Hallquist et al., 2009; Rantala et al., 2015). However, long-term records indicate that BVOC emissions at this site have remained relatively stable over the past two decades (Kulmala et al., 2001). As such, no evidence currently supports a significant contribution of BVOC variability to the observed multi-year decline in PM<sub>1</sub> scattering. Taken together, the trend observed at

- SMEAR II is consistent with known reductions in anthropogenic precursor emissions, particularly SO<sub>2</sub>, though
- additional factors cannot be excluded.
- 3.2.2 Light absorption coefficient at 520 nm ( $\sigma_{abs, 520}$ )
- For PM<sub>10</sub> and PM<sub>1</sub>,  $\sigma_{abs,520}$  decreases significantly over the record. PM<sub>10</sub> declines at  $-0.11 \pm 0.03$  Mm<sup>-1</sup> yr<sup>-1</sup> (-8.75
- $\pm$  2.11 % yr<sup>-1</sup>; Fig. S1b; Table S3b), and PM<sub>1</sub> at  $-0.09 \pm 0.02$  Mm<sup>-1</sup> yr<sup>-1</sup> ( $-8.87 \pm 2.18$  % yr<sup>-1</sup>; Fig. S2b; Table
- S4b). Both series exhibit winter maxima and a recurring late-spring minimum, in agreement with the established
- seasonal cycle at SMEAR II (Luoma et al., 2019) and consistent with Europe-wide declines in aerosol absorption
- and black-carbon emissions (Collaud Coen et al., 2020; Yttri et al., 2021).
- The close agreement of the relative declines indicates that the PM<sub>10</sub> absorption trend is likely dominated by the
- fine-mode (PM<sub>1</sub>) contribution at 520 nm, based on the corrected and harmonized time series described in Section
- 2.3. For physical context—not used in estimating the trends reported above—coarse-mode particles typically
- contain weakly absorbing mineral and biological components (Laskin et al., 2005; Moosmüller et al., 2009) and
- have shorter atmospheric residence times due to gravitational settling (Emerson et al., 2020; Zhang et al., 2001).
- Considerations related to long-term sensitivity, harmonization, and instrument intercomparison are documented
- elsewhere (Collaud Coen et al., 2010) and do not affect the magnitude or direction of the PM10/PM1 trends
- summarized here.

387

- 3.2.3 PM mass concentration (*PM mass*)
- Although this study focuses on  $PM_1$  and  $PM_{1-10}$  fractions, investigating trends in super- $PM_{10}$  aerosol particles may
- provide additional insights into long-term shifts in coarse-mode aerosol composition that are not detected by the
- optical instruments and the used 10 μm cut-off. Super-PM<sub>10</sub> particles, which include biological aerosols (e.g.,
- pollen, fungal spores) and mineral dust, are strongly influenced by episodic events such as seasonally driven
- biological emissions and long-range dust transport. However, their long-term trends remain poorly constrained.
- The variability in super-PM<sub>10</sub> aerosols likely reflects a combination of natural and anthropogenic influences.
- While mineral dust and pollen contribute significantly to coarse-mode particles, their variability is largely seasonal
- and event-driven. In contrast, fine-mode aerosol particles (PM<sub>1</sub>) and PM<sub>1-10</sub> aerosol particles exhibit more
- consistent long-term trends due to anthropogenic emissions. The decline in PM<sub>1</sub> mass concentrations aligns with
- reductions in both primary and secondary anthropogenic aerosol sources. However,  $PM_{1-10}$  contains a larger
- fraction of natural aerosols, which exhibit substantial variability but may not necessarily follow a long-term trend.
- Previous studies (Leskinen et al., 2012; Varga et al., 2023) have highlighted the episodic nature of coarse-mode
- aerosol contributions, including periods of increased dust transport to Finland, particularly after 2010 (Varga et
- al., 2023). The frequency of these events, combined with changes in regional meteorology and source
- contributions, may introduce variability in observed  $PM_{1-10}$  trends. In this study, we further assess the relative
- contributions of biological aerosols and mineral dust by comparing super-PM<sub>10</sub> variations with PM<sub>1-10</sub>, providing
- insights into their role in long-term aerosol trends and atmospheric transport patterns.

## 3.3 Long-term trends of intensive properties

**Figure 2.** Time series of (a) AAE, (b) SAE, (c)  $SSA_{550}$ , (d)  $MSC_{550}$  and (e)  $MAC_{520}$  for the  $PM_{1-10}$  aerosol particles from October 2010 to October 2022. The blue shaded area is the interquartile range (25th–75th percentile) of monthly values; the blue line is the monthly median. The red line shows the Theil-Sen trend (solid if p  $\leq$  0.05; dashed otherwise). Months with <75% data coverage are left blank.

# 3.3.1 Absorption Ångström exponent (AAE)

The AAE for  $PM_{1-10}$  aerosol particles exhibits a statistically significant decreasing trend with a slope of  $0.03 \pm 8.63 \times 10^{-3} \, \text{yr}^{-1}$ ;  $3.54 \pm 1.04 \, \text{\%} \, \text{yr}^{-1}$  (Figure 2(a); Table 3(d)), suggesting a shift in aerosol composition. The more pronounced decrease in  $PM_{1-10}$  suggests a potential decline in BrC, possibly due to enhanced oxidation processes or a shift toward less absorbing organic aerosols. This aligns with long-term reductions in BC concentrations at Hyytiälä (Luoma et al., 2021) and shifts in organic aerosol sources (Äijälä et al., 2019; Heikkinen et al., 2020). Given that BC is predominantly submicron, the decreasing AAE in  $PM_{1-10}$  is more likely to reflect changes in BrC absorption characteristics in the coarse-mode fraction rather than BC variability alone.

A notable abrupt change in *AAE* values is observed between pre-2018 and post-2018, which may be linked to the instrumental transition from AE31 to AE33 in March 2018. The AE33 aethalometer introduces real-time filter-loading corrections through a dual-spot measurement approach, which effectively reduces filter-loading biases inherent to the AE31 model (Drinovec et al., 2015). This methodological difference has been shown to impact BC and BrC estimates, particularly influencing *AAE* calculations (Backman et al., 2017; Zotter et al., 2017). Also, Luoma et al. (2021) showed how *AAE* varies between different aethalometer correction algorithms. Interestingly, this discontinuity is not observed in PM<sub>1</sub> or PM<sub>10</sub> *AAE* trends, which suggests that the AE31-to-AE33 transition might have introduced size-dependent effects in *AAE* calculations. Previous studies indicate that AE33 generally yields lower *AAE* values than AE31, especially at shorter wavelengths where BrC absorption dominates, which could affect how BrC in coarse-mode aerosols (PM<sub>1-10</sub>) is quantified (Bernardoni et al., 2021; Zotter et al., 2017).

While these differences align with known AE31-to-AE33 biases, further analysis is needed to determine whether the observed *AAE* shift stems solely from instrumental changes or also reflects atmospheric variations. Examining

the AAE wavelength dependence across size fractions and assessing potential modifications in BrC optical properties could help separate instrumental artifacts from real aerosol composition trends. At least one previous study, Bali et al. (2024), highlights the importance of BrC optical properties in interpreting long-term aerosol trends, particularly in environments influenced by SOA and episodic biomass-burning events.

## 3.3.2 Scattering Ångström exponent (SAE)

The *SAE* for PM<sub>1-10</sub> particles exhibits a negative trend of  $-0.03 \pm 0.02$  yr<sup>-1</sup>;  $-7.30 \pm 4.16$  %yr<sup>-1</sup> (Figure 2(b); Table 3(c)), indicating a declining influence of smaller particles (closer to 1  $\mu$ m) or a slower decline of larger particles (closer to 10  $\mu$ m), given concurrent decreases in  $\sigma_{sea,550}$  and *PM mass*. This suggests that fine-mode aerosols within PM<sub>1-10</sub> are decreasing at a faster rate than those in the 2.5–10  $\mu$ m range, leading to a relative dominance of larger particles rather than an absolute increase. The observed trend may be driven by reduced secondary aerosol formation or changes in atmospheric processing (Maso et al., 2005; Seinfeld & Pandis, 2016). A decline in fine-mode SOA condensation onto pre-existing particles could limit fine-mode growth, shifting the relative contribution of supermicron particles (>1  $\mu$ m). Additionally, long-term reductions in sulfate emissions make it unlikely that an increase in sulfate mass is responsible for the shift in *SAE*. Instead, changes in dust concentrations could play a role, although long-term dust trends remain uncertain. The negative *SAE* trend observed for PM<sub>10</sub> (Figure S3(b)) further suggests a greater relative contribution of larger particles within PM<sub>10</sub>, likely due to a slower decline in the 2.5–10  $\mu$ m range compared to fine-mode particles (1–2.5  $\mu$ m). Aerosol particles in this size range, including mineral dust, sea salt, pollen and fungal spores, exhibit different atmospheric lifetimes and removal processes than finer particles, contributing to differences in observed trends.

For PM<sub>1</sub> (Figure S4(b)), SAE remains stable, suggesting no major shifts in fine-mode size distributions. This stability indicates that the ratio of BC to scattering components, primarily sulfate and organics, has not changed significantly over time. Luoma et al. (2019) reported a decrease in volume mean diameter (VMD) from ~0.25  $\mu$ m to ~0.2  $\mu$ m (2006–2017), indicating a long-term shift toward smaller aerosol sizes. While this period does not fully overlap with this study, the trend likely continued, though more recent data would be needed to confirm this assumption. The observed differences between PM<sub>1</sub> and PM<sub>1-10</sub> SAE trends highlight the distinct processes influencing fine and larger aerosols, including atmospheric processing, long-range transport, and primary emissions. Reduced SOA condensation onto fine particles may further enhance the relative contribution of larger particles in PM<sub>10</sub>. However, while pollen (>10  $\mu$ m) is classified as a coarse aerosol in this study, its influence on SAE trends is likely minimal, as infrequent high concentrations from pollen events are statistically down-weighted in the calculation of monthly medians. The concurrent decline in  $\sigma_{sca,550}$  and PM mass indicates that the negative SAE trend is not due to an absolute increase in large particles but rather a differential reduction in size fractions. Together, these findings suggest that ongoing changes in secondary aerosol formation and primary emissions are influencing the evolution of aerosol size distributions in different size fractions.

#### 3.3.3 Single scattering albedo (SSA550)

The SSA<sub>550</sub> for PM<sub>10</sub>, PM<sub>1</sub>, and PM<sub>1-10</sub> shows no statistically significant long-term trends over the measurement period (Figures 1(c), 2(c), 3(c); Table 3(f)). Statistical significance was evaluated using the Mann–Kendall test (*p* compared with 0.05) and Theil–Sen slopes with 95 % confidence intervals. Although the fitted slopes are positive

across all size fractions, they are not significant and cannot be interpreted as evidence of systematic long-term changes in the balance between scattering and absorbing components. The lack of statistically significant trends likely reflects the relatively short record (~2010–2022), the high natural variability of SSA550 driven by meteorology, regional and long-range transport, and episodic contributions from pollen and mineral dust. These influences are particularly strong in the coarse and supermicron fractions, where sporadic sources dominate, limiting the statistical power to resolve weak tendencies. The absence of concurrent long-term chemical composition and source-apportionment datasets further constrains attribution of *SSA550* variability to specific aerosol species or emission changes.

Changes in *SSA* are generally associated with reductions in absorbing components, such as black carbon, or increases in non-absorbing species, such as sulfates and organics (Bond et al., 2013; Luoma et al., 2019; Pandolfi et al., 2014). However, regional SO<sub>2</sub> emissions have declined substantially over recent decades (Klimont et al., 2013), making a sulfate-driven increase in *SSA*<sub>550</sub> at Hyytiälä unlikely. Long-term measurements at the site show that organic aerosol dominates the aerosol mass fraction, with variability largely seasonal rather than indicative of sustained long-term changes (Heikkinen et al., 2021; Äijälä et al., 2019). Without detailed separation of primary and secondary organic aerosol and the absence of continuous long-term dust measurements, it remains difficult to evaluate whether any weak tendencies toward increased scattering are linked to changes in mineral dust or other primary sources. Overall, the *SSA*<sub>550</sub> observations indicate that the relative balance between scattering and absorbing aerosol components has remained broadly stable at Hyytiälä over the past decade.

## **3.3.4** Mass scattering coefficient (*MSC*<sub>550</sub>)

From October 2010 to October 2022, the PM<sub>1-10</sub>  $MSC_{550}$  time series is episodic but stable in the long term. Monthly medians broaden in late spring and summer when pollen and dust are frequent, yet the central tendency remains flat; the fitted trend is not different from zero (slope:  $-0.02 \pm 0.01$  m<sup>2</sup>g<sup>-1</sup>yr<sup>-1</sup>; relative trend:  $-1.13 \pm 0.94$  %yr<sup>-1</sup>; Fig. 2(d); Table 3). A recurrent winter-high/summer-low seasonal cycle persists throughout the record and no step change is evident across the 2018 instrument transition. Event—non-event differences in distribution are confirmed with a one-sided Mann—Whitney U test (Section 4.2), indicating that episodes widen the spread but do not impose a drift. For context, PM<sub>10</sub> and PM<sub>1</sub> show larger  $MSC_{550}$  magnitudes yet the same time-series behaviour, that is, short-term variability around a level baseline, with trends statistically indistinguishable from 0 (PM<sub>10</sub> slope: 3.70 x  $10^{-3} \pm 0.02$  m<sup>2</sup>g<sup>-1</sup>yr<sup>-1</sup>; PM<sub>10</sub> relative trend:  $0.15 \pm 0.64$  %yr<sup>-1</sup>; PM<sub>1</sub> slope:  $1.54 \times 10^{-3} \pm 0.02$  m<sup>2</sup>g<sup>-1</sup>yr<sup>-1</sup>; PM<sub>1</sub> relative trend:  $0.05 \pm 0.54$  %yr<sup>-1</sup>; Figures. S3(d) and S4(d); Tables S3(g) and S4(g)).

#### 3.3.5 Mass absorption coefficient ( $MAC_{520}$ )

For PM<sub>1-10</sub>,  $MAC_{520}$  decreases significantly (slope:  $-2.77 \times 10^{-3} \pm 9.04 \times 10^{-4}$  m<sup>2</sup>g<sup>-1</sup>yr<sup>-1</sup>; relative trend:  $-3.97 \pm 1.30$  % yr<sup>-1</sup>; Figure 4e; Table 3(h)). PM<sub>10</sub> and PM<sub>1</sub> exhibit no significant trends (PM<sub>10</sub>:  $-9.30 \times 10^{-3} \pm 3.29 \times 10^{-3}$  482 m<sup>2</sup>g<sup>-1</sup>yr<sup>-1</sup>, Table S3(h); PM<sub>1</sub>:  $-0.01 \pm 5.08 \times 10^{-3}$  m<sup>2</sup>g<sup>-1</sup>yr<sup>-1</sup>, Table S4(h)). Values use the pseudo-daily means defined in Section 2.3. In Figure S10, PM<sub>10</sub> is shown in panels (a) AE31 (2010–2017) and (b) AE33 (2018–2022), PM<sub>1</sub> in (c) AE31 (2010–2017) and (d) AE33 (2018–2022), and PM<sub>1-10</sub> in (e) AE31 (2010–2017) and (f) AE33 (2018–2022). A positive instrument-related offset is evident for AE33 relative to AE31, which is most pronounced

for the  $PM_{10}$  and  $PM_{1-10}$  aerosol particles and is consistent with reduced filter-loading bias due to the AE33 dual-spot correction. This positive offset makes any 2010–2022 decrease appear smaller, so the observed  $PM_{1-10}$  decline in the  $MAC_{520}$  is not caused by the instrument change (Bond et al., 1999; Weingartner et al., 2003; Virkkula, 2010; Drinovec et al., 2015; Zotter et al., 2017).

# 3.4 Seasonal variability of the extensive properties

The  $\sigma_{abs,520}$  follows a strong seasonal cycle, with the highest values observed in winter (December, January and February) due to increased BC emissions from residential wood burning (Kukkonen et al., 2020; Pandolfi et al., 2014) in Finland. Additionally, the lower PBL during winter reduces vertical dispersion, leading to higher near-surface aerosol concentrations due to thermal inversion (Petäjä et al., 2016). These findings are consistent with Hyvärinen et al. (2011), which reported elevated BC concentrations in Finland during winter, primarily due to increased heating emissions and stable atmospheric conditions. In contrast,  $\sigma_{abs,520}$  values are lower in May, June and July, mainly due to reduced heating emissions and an elevated PBL, which enhances aerosol dispersion (Arola et al., 2011). It is also possible that the correction algorithms were not entirely sufficient to minimize the sensitivity of the absorption measurements to the accumulation of aerosol particles on the filter. Meanwhile, the  $\sigma_{sca,550}$  for PM<sub>10</sub>, PM<sub>1</sub>, and PM<sub>1-10</sub> also exhibits distinct seasonal variability, with the highest values occurring in summer (June, July and August) and the lowest in April and October (Figures S5(a), S6(a) and 3(a)). The summer peak is attributed to increased emissions of BVOCs and enhanced SOA formation, consistent with findings that SOA dominates aerosol composition in boreal regions during summer (Kourtchev et al., 2016; Tunved et al., 2006).

**Figure 3.** Monthly variations of aerosol optical properties for the  $PM_{1-10}$  aerosol particles from October 2010 to October 2022 (a)  $\sigma_{sca,550}$ , (b)  $\sigma_{abs,520}$ , and (c) PM mass. Box plots represent the interquartile range, with the first horizontal line showing the  $25^{th}$  percentile, the middle line showing the median ( $50^{th}$  percentile), and the third horizontal line showing the  $75^{th}$  percentile. Whiskers extend to the  $10^{th}$  and  $90^{th}$  percentiles.

The April and October minima are linked to reduced household heating emissions in spring and their delayed onset in autumn, along with weaker long-range transport contributions during these transitional months (Hienola et al., 2013). The seasonal variation in PM mass at Hyytiälä, as shown in Figure S5(c) for PM<sub>10</sub>, Figure S6(c) for PM<sub>1</sub> and Figure 3(c) for PM<sub>1-10</sub> aerosol particles, further highlights differences between aerosol size fractions. PM<sub>1</sub> mass concentrations increase significantly during June–August, coinciding with enhanced SOA production, as indicated by the concurrent rise in  $\sigma_{sca,550}$  for PM<sub>1</sub> Figures S6(a). These trends align with long-term aerosol observations in Finland (Luoma et al., 2019), which report similar seasonal variations in PM mass and composition. Meanwhile, PM<sub>10</sub> and PM<sub>1-10</sub> mass also show seasonal variability, increasing in May, coinciding with the peak in birch pollen emissions (Yli-Panula et al., 2009). Additionally, June and July peaks in these fractions may indicate long-range transported mineral dust from the Aral-Caspian and Middle Eastern regions (Varga et al., 2023). Seasonal variations in atmospheric transport and aerosol sources further contribute to these trends, with summer months favoring photochemical activity and secondary aerosol formation, while winter remains dominated by local primary emissions.

## 3.5 Seasonal variability of the intensive properties

The seasonal variability of intensive aerosol properties at Hyytiälä provides insight into the sources and composition of PM<sub>10</sub> and PM<sub>1-10</sub> aerosol particles. Figures 4(d) and 4(e) illustrate the monthly variations of the  $MAC_{520}$  and the  $MSC_{550}$ .  $MAC_{520}$  peaks in winter due to increased emissions from biomass burning and residential heating, as shown by Heikkinen et al. (2021) and Virkkula et al. (2011). Reduced boundary-layer heights further enhance BC accumulation, leading to higher  $MAC_{520}$  values. Conversely, the  $MAC_{520}$  declines in summer due to the dominance of SOA from biogenic sources (Äijälä et al., 2019). Organic aerosols scatter more light than they absorb, reducing the  $MAC_{520}$  values. The seasonal minimum in the imaginary part of the refractive index further indicates lower absorption capacity in summer, reinforcing the shift toward scattering-dominated aerosols (Virkkula et al., 2011).

The  $MSC_{550}$  follows a distinct seasonal cycle, with peak values in winter (Figure 4(d)). The prevalence of fine-mode PM<sub>1</sub> particles, primarily from residential heating, enhances light scattering (Hellén et al., 2008). Although BC is present, the aerosol mixture includes significant amounts of scattering species such as SOA and sulfates, leading to high  $MSC_{550}$  values (Seinfeld & Pandis, 2016). Despite an increase in the PM mass in summer,  $MSC_{550}$  declines due to a shift toward larger particles that scatter light less efficiently per unit mass (Tunved et al., 2006). The seasonal reduction in the  $MSC_{550}$  aligns with greater contributions from pollen, mineral dust, and sea salt aerosols in the  $PM_{1-10}$  fraction, altering the aerosol size distribution (Latimer & Martin, 2019). These shifts emphasize the importance of size-dependent scattering efficiency in shaping seasonal aerosol optical properties.

**Figure 4.** Monthly variations of aerosol optical properties for the  $PM_{1-10}$  aerosol particles from October 2010 to October 2022 (a) AAE, (b) SAE, (c) SSA<sub>550</sub>, (d) MSC<sub>550</sub> and (e) MAC<sub>520</sub>. Box plots represent the interquartile range, with the first horizontal line showing the 25<sup>th</sup> percentile, the middle line showing the median (50<sup>th</sup> percentile), and the third horizontal line showing the 75<sup>th</sup> percentile. Whiskers extend to the 10<sup>th</sup> and 90<sup>th</sup> percentiles.

**Figure 5.** Temporal trend comparisons of PM mass in (a)  $PM_{10}$ , (b) super- $PM_{10}$  aerosol particles and seasonal variation comparisons of  $PM_{10}$  and super- $PM_{10}$  aerosol particles in terms of (c) PM mass concentration and (d) PM mass fraction, (e) number of dust and pollen events (counts) and (f) number of dust and pollen events (%).

The  $PM_{1-10}$  fraction significantly influences seasonal aerosol dynamics, particularly in spring and summer when pollen and mineral dust events contribute to coarse-mode aerosol mass. However, optical measurements in this study are constrained by the  $PM_{10}$  inlet, excluding particles larger than 10  $\mu$ m. Figure 5 quantifies this missing PM fraction, showing substantial coarse-mode mass is unaccounted for, especially during episodic events. The seasonal increase in  $PM_{1-10}$  and  $PM_{10}$  mass suggests coarse-mode aerosols dominate specific periods, yet their optical contributions remain uncertain. Complementary measurement techniques are needed to capture the full size distribution of coarse-mode particles and improve the representation of their optical properties. The exclusion of super- $PM_{10}$  from optical instruments introduces uncertainties in radiative forcing assessments, underscoring the necessity of including larger particles in aerosol characterization.

Pseudo-daily peaks in super-PM<sub>10</sub> mass (Figure 5) indicate that episodic events, such as pollen and dust outbreaks, significantly impact aerosol optical properties. These events contribute to short-term fluctuations in the *MAC*<sub>520</sub> and the *MSC*<sub>550</sub>, reflecting shifts in aerosol composition and size distribution. The variability of super-PM<sub>10</sub> mass suggests coarse-mode particles play a crucial role in modifying scattering and absorption properties, further complicating aerosol–radiation interactions. Without direct optical measurements of these particles, their contribution to aerosol optical closure remains uncertain. The high mass fraction of super-PM<sub>10</sub> during episodic events suggests that excluding these particles leads to an underestimation of coarse-mode aerosol influences in climate models. The seasonal trends and episodic peaks observed in Figures 4 and 5, respectively emphasize the need for improved measurement techniques and better parameterization of coarse-mode aerosols in radiative forcing assessments.

## 4. Role of episodic variability

#### 4.1 Optical and mass properties of PM<sub>10</sub> aerosol particles at Hyytiälä: Episodic events and long-term trends

The optical and mass properties of  $PM_{10}$  aerosol particles at Hyytiälä exhibit variability influenced by episodic events and long-term trends. To classify the dominant aerosol types, we applied the framework by Cazorla et al. (2013), which utilizes the SAE and the AAE, as shown in Figure 6.

Figure 6. Aerosol classification matrix for the PM<sub>10</sub> aerosol particles.

582

587

594

595

Figure 6 presents the aerosol classification for PM<sub>10</sub> particles, as PM<sub>10</sub> encompasses both PM<sub>1</sub> and PM<sub>1-10</sub> aerosols, providing a broader representation of aerosol interactions. The data show that PM<sub>10</sub> particles predominantly fall within the 'EC/OC-dominated' and 'Dust/EC-dominated' regions, indicating contributions from combustionderived BC, SOA, and aged dust.

Pollen and dust events appear sporadically, clustering in moderate to high AAE and SAE regions, reinforcing their episodic nature. These events introduce short-term fluctuations in aerosol properties but do not significantly alter the long-term classification. Dust events cluster near the 'Dust/EC-dominated' region, suggesting interactions between long-range transported dust and combustion aerosols. Pollen events align with higher SAE (~1.5-2), indicating smaller particles with strong scattering properties.

Despite episodic variability, the dominant aerosol classification remains stable, with PM<sub>10</sub> aerosols primarily linked to EC/OC sources and aged dust.

- Table S3 summarizes the classification scheme used in this study, differentiating aerosol types based on their optical properties:
- (1) High SAE (>1.5) and low AAE (~1): BC-dominated aerosols from fossil fuel combustion and biomass burning.
- (2) High AAE (>2) and low SAE (<1): Mineral dust, often associated with long-range transport.
  - (3) Intermediate AAE (1-2) and SAE (~1): Aged dust mixed with BC, indicative of interactions between transported dust and combustion aerosols.

#### 4.2 Episodic and long-term variability: One-sided Mann-Whitney U test

To assess the impact of episodic events (i.e., pollen and dust) on aerosol properties, a one-sided Mann-Whitney U test was conducted. This non-parametric statistical test is particularly suited for comparing two independent datasets without assuming normality, making it effective for aerosol optical property distributions, which often exhibit non-Gaussian behavior due to episodic influences.

The dataset was categorized into:

- (a) Observations including pollen and/or dust events
- (b) Observations excluding these events

Pollen events were identified using cascade impactor filter records. If the  $PM_{10}$  filters contained pollen, the corresponding  $PM_{1-10}$  fraction was also assumed to contain pollen.

Dust events were identified based on time periods from Varga et al. (2023), cross-referenced with aerosol optical and mass data from Hyytiälä.

Table 4. One-sided Mann-Whitney U-test results for the aerosol optical and mass properties in the presence of pollen and/or dust events for the  $PM_{1-10}$  aerosol particles

| Variable           | Number of<br>data points<br>(pollen<br>and/or dust<br>events) | Number of<br>data points<br>(excluding<br>pollen and<br>/or dust<br>events) | U-statistic | p-value                  | Statistical<br>significance | Trend      |
|--------------------|---------------------------------------------------------------|-----------------------------------------------------------------------------|-------------|--------------------------|-----------------------------|------------|
| $\sigma_{abs,520}$ | 23                                                            | 1208                                                                        | 17442       | 0.04                     | Yes                         | Decreasing |
| $\sigma_{sca,550}$ | 28                                                            | 1446                                                                        | 28653       | 1.64 x 10 <sup>-4</sup>  | Yes                         | Decreasing |
| PM mass            | 30                                                            | 1533                                                                        | 39156       | 4.08 x 10 <sup>-11</sup> | Yes                         | Decreasing |
| AAE                | 23                                                            | 1182                                                                        | 14692       | 0.51                     | No                          | No trend   |
| SAE                | 28                                                            | 1441                                                                        | 21013       | 0.71                     | No                          | No trend   |
| SSA <sub>550</sub> | 21                                                            | 1172                                                                        | 12977       | 0.67                     | No                          | No trend   |
| $MAC_{520}$        | 23                                                            | 1179                                                                        | 6537        | 2.06 x 10 <sup>-5</sup>  | Yes                         | Decreasing |
| MSC <sub>550</sub> | 28                                                            | 1413                                                                        | 10798       | 3.79 x 10 <sup>-5</sup>  | Yes                         | Decreasing |

- A Mann–Whitney U test (one-sided) shows that  $\sigma_{sca,550}$  (n=28 pseudo-daily means) and PM mass (n=30) are
- higher during pollen/dust events than during non-event periods (event median > non-event median; p-value ≤
- 0.05).  $MAC_{520}$  (n = 23) is lower during events (event median < non-event median; p-value  $\le 0.05$ );  $MSC_{550}$  (n = 23)
- 28) also differs significantly, and  $\sigma_{abs,520}$  (n=23) shows a weaker but significant difference.  $SSA_{550}$  (n=21), AAE
- (n = 23), and SAE (n = 28) show no significant differences.  $PM_{10}$  mostly scatters light, but mineral dust in this
- size range can also absorb (Adebiyi et al., 2023).
- Trends were estimated with a modified Theil–Sen slope and significance was tested with the Hamed & Rao (1998)
- autocorrelation-corrected Mann–Kendall test. The  $MAC_{520}$  decreases significantly (p-value  $\leq 0.05$ ), while  $SSA_{550}$ ,
- AAE and SAE show no trend (p-value > 0.05); no other trends are claimed.  $\sigma_{sca}$  has been corrected for angular
- truncation and non-ideal angular response; some uncertainty remains because the correction depends on particle
- size distribution and refractive index (Anderson & Ogren, 1998; Müller et al., 2011). Filter-based absorption can
- retain residual multiple-scattering and loading artifacts that bias  $\sigma_{abs}$  high (and thus MAC) (Weingartner et al.,
- 2003; Bond et al., 1999; Ogren, 2010; Virkkula, 2010).
- We show  $PM_{10}$  because it includes both  $PM_1$  and  $PM_{1-10}$  and the optics use a  $PM_{10}$  inlet (particles > 10  $\mu$ m
- excluded). The data cluster mainly in the EC/OC mixture and Dust/EC mix regions, with few points in Dust
- dominated and OC/Dust mix. We therefore base interpretation on the better-sampled mixed regimes and separate
- episodic events from longer-term behavior (Adebiyi et al., 2023; Che et al., 2018), consistent with boreal
- observations of SOA-dominated backgrounds and mixed BC-organic conditions (Virkkula et al., 2011; Hyvärinen
- et al., 2011).

## 5. Summary and conclusions

- This study provides new insights into the long-term evolution of aerosol optical properties and PM mass at the
- SMEAR II station in Hyytiälä, southern Finland, over the past 12 years. A significant decrease in extensive aerosol
- properties in PM<sub>10</sub> aerosols suggests a shift in size distribution and chemical composition, likely driven by
- declining anthropogenic emissions, including SO<sub>2</sub>, across northern Europe (Tørseth et al., 2012; Zieger et al.,
- 2010). Seasonal patterns show that absorbing aerosols peak in winter due to biomass burning and residential
- heating, while scattering aerosols peak in summer, influenced by biogenic SOA formation and episodic pollen
- and dust events. These findings reinforce previous research on boreal aerosol processes and highlight the complex
- interactions between anthropogenic and natural aerosol sources at this site.
- To address the challenge of incomplete optical closure, this study examines the role of supermicron aerosol
- particles across multiple seasons. The first-ever quantification of super-PM<sub>10</sub> particles at Hyytiälä underscores
- their role in aerosol scattering and absorption processes, contributing to uncertainties in aerosol-radiation
- interactions. Statistical analysis revealed that pollen and dust events significantly affected five of the eight  $PM_{1-10}$
- aerosol optical and mass-related properties examined. The observed trends in  $MAC_{520}$  and  $MSC_{550}$  suggest changes
- in aerosol composition and size distribution, which must be considered in radiative forcing assessments. Future
- work should improve the representation of supermicron aerosols in climate models by leveraging advanced
- analytical techniques, such as machine learning-based aerosol classification (Schuster et al., 2006) and integrating
- multi-platform observational datasets to reduce uncertainties.

#### 648 Code/Data availability

- The analysis codes are available in the Zenodo repository <a href="https://doi.org/10.5281/zenodo.17393221">https://doi.org/10.5281/zenodo.17393221</a>, and the
- processed data are available in a separate Zenodo repository <a href="https://doi.org/10.5281/zenodo.17391504">https://doi.org/10.5281/zenodo.17391504</a> (Banerji et
- al., 2025).

#### **Author contributions**

- SB, KL and TP conceptualized the study. SB performed the data analysis. IY and LA contributed to the collection
- of data at SMEAR II. SB wrote the original draft and prepared the manuscript with input from KL. KL, IY, VMK
- and TP contributed to the review and revision of the manuscript, including the final edits. TP provided overall
- supervision and secured funding.

# 657 Competing interests

- One of the co-authors is a member of the editorial board of Atmospheric Chemistry and Physics. This co-author
- was not involved in the peer-review or editorial decision-making process for this manuscript. The authors declare
- no other competing interests.

#### 661 Acknowledgements

- The funding from ACTRIS-Finland host organizations University of Helsinki (UH) and INAR RI/ ACTRIS-FI
- 2020-2024 grant no. 328616, INAR RI 2022-2025 grant no. 345510 (UH), and the ACCC (Atmosphere and
- Climate Competence Center) Flagship funding by the Research Council of Finland (grant no. 337549 (UH) are
- gratefully acknowledged.
- Support of the European Commission via non-CO<sub>2</sub> forcers and their climate, weather, air quality and health impacts
- (FOCI, project number 101056783) is gratefully acknowledged.
- During the preparation of this work the authors used ChatGPT to improve the readability of certain sections. After
- using this tool, the authors reviewed and edited the content as needed and take full responsibility for the content of
- the manuscript.

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
