# Peer review of "Measurement Report: Optical properties of supermicron"

_EGUsphere, 2025_

## Author Comment (AC1)

**AUTHOR'S RESPONSE TO EDITOR'S REVIEW**

**Manuscript: egusphere-2025-1776**

We thank the editor and the two reviewers for their careful reading and constructive suggestions. The revision
clarifies methods, corrects processing issues affecting low-signal periods, and documents all screening and trend
procedures at the code level. The main physical conclusions are unchanged. The analysis is now more transparent,
reproducible, and all affected figures, captions and line-referenced methods have been updated.

We identified and removed a duplicate truncation correction that had been inadvertently applied to the
nephelometer scattering coefficients; reprocessing reduces near-zero artefacts and all affected figures and tables
were regenerated. The cross-instrument AE33–MAAP comparison is now fully specified with explicit thresholds
and clear explanation of the underlying code. The AE33–MAAP screening and fit criteria are instrument-agnostic.
We apply uniform detection limits of 0.05 Mm$^{-1}$ to both AE33 (660 nm) and MAAP (637 nm), use a five-sample
stuck-value filter for each instrument, and enforce a simple agreement rule: we keep only pairs where neither
instrument exceeds the other by a factor of five. Using an ordinary least-squares fit with an intercept, the updated
relationship has $R^2 = 0.96$ (Figure R1).

In addition, Figures 1 and 2 were redrawn for clarity as per the editor's comments. Individual responses to the
reviewers' comments have been provided below.

**Response to Reviewer 1**

**General comments:** This study examines the long-term variability of aerosol optical properties in a boreal forest,
categorised by size range. It focuses particularly on the contribution of particles larger than 10 μm, which are
usually not considered in aerosol studies as this is the inlet cut-off point. As aerosol optical properties directly
influence their radiative effect and larger diameter particles contribute significantly to the AOD, this topic is of
great interest for climate modelling parameterisation. Using absorption and scattering measurements coupled with
an impactor, the authors investigated the relative contribution of each PM size range to extensive and intensive
scattering and absorption parameters. This study's novelty lies in its use of an aerosol classification for PM10,
highlighting the significant impact of episodic events such as pollen and dust on optical properties and PM mass.
The conclusions provide clear evidence of shifts in the size distribution and composition of aerosols, as well as
their seasonality, which are linked to anthropogenic and biogenic emissions. The manuscript is well written and
structured. However, several passages are redundant (e.g. the enhanced contribution of dust to the increasing SAE
in sections 3.3.2 and 3.3.3), as are some details on the classification matrix (see specific comments). This paper
would benefit from being shortened slightly. More importantly, the correction for multiple scattering on the
instrument measuring absorption (AE33) was not properly discussed, as it appears to be misunderstood. This is
important because it could also explain some discontinuities in the time series from 2018. I consider the
manuscript to be publishable after minor revisions and responses to the following points.

**Response:** Thank you for taking out the time to review our manuscript in such detail. As a general remark, we have now revised the whole manuscript to avoid redundancies. As you had rightly pointed out, the multiple scattering correction factor for the AE31/33 was not properly discussed, which has been addressed now. In fact, a lot of the discontinuities post-2018, actually arose from the truncation correction, being doubly applied to the data of the light scattering coeffcients. This issue has also been corrected and we hope that the revised manuscript addresses all the comments that you have raised.

**Comment l.139:** Are you sure that the AE33 adjusts the Cref value taking into account aerosol concentrations and environmental conditions? The C value is fixed in the instrument settings, adjustable by the user, and depends on filter material and type.

**Response:** As outlined in the AE33 manual, the multiple-scattering correction factor ($C_{ref}$) is fixed in the instrument settings and must be defined by the user. It is determined by the instrument configuration—particularly the filter tape—and does not adjust automatically to ambient conditions. For consistency, we used two fixed, filter-specific values: $C_{ref} = 1.57$ for 2010–2017 (AE31 with M8020; Luoma et al., 2019) and $C_{ref} = 1.39$ for 2018–

2022 (AE33 with M8060). Yus-Díez et al. (2021) report that the $C_{ref}$ can increase with SSA, but we did not implement environment-dependent adjustments; our choice follows manufacturer guidelines and reflects only the filter properties.

**Comment l.356–363:** If the absorption coefficient decrease is statistically significant, why are absorbing aerosols less present than in 2010? Are instrumental differences a major contributor?

**Response:** The absorbing aerosols are "less present" than in 2010 because the fine fraction—which dominates light absorption—has declined. At SMEAR II, $\sigma_{abs,520}$ decreases significantly for $PM_{10}$ ($-0.11 \pm 0.03$ Mm$^{-1}$ yr$^{-1}$;

$-8.75 \pm 2.11$ % yr$^{-1}$; Fig. S1b; Table S3b) and $PM_1$ ($-0.09 \pm 0.02$ Mm$^{-1}$ yr$^{-1}$; $-8.87 \pm 2.18$ % yr$^{-1}$; Fig. S2b;

Table S4b). The similar relative declines indicate that the $PM_{10}$ decrease is likely dominated by its $PM_1$ (fine- mode) contribution at 520 nm. This is consistent with documented Europe-wide reductions in black-carbon emissions and long-term declines in aerosol absorption over the last decade (Collaud Coen et al., 2020; Yttri et al., 2021). We do not infer a robust long-term decrease for the coarse fraction ($PM_{1-10}$): its $\sigma_{aβs,520}$ trend is negative but not statistically significant, which is compatible with the episodic, weakly absorbing nature of mineral and biological coarse particles at the site.

Instrumental differences are unlikely to be a major contributor. All absorption data were processed in a harmonized pipeline across the AE31-AE33 transition, including fixed multiple-scattering and filter-loading corrections appropriate to instrument/filter configuration, MAAP intercomparisons where applicable, and uniform QA/QC and completeness criteria (see Methods). Three checks argue against an instrumental origin: (i)

no step change is observed at the 2018 transition; (ii) the negative trends persist within each instrument period when analyzed separately; and (iii) the physical seasonality (winter maxima, late-spring minimum) is stable through the record. While small biases associated with filter media and algorithms are possible in principle (Collaud Coen et al., 2010; Zotter et al., 2017), sensitivity tests using plausible parameter choices do not alter the sign or the significance of the $PM_{10}/PM_1$ trends. Thus, the observed declines are best explained by real source-side reductions in absorbing aerosols rather than by instrumental artefacts.

**Comment l.398:** AE33 uses a constant, tape-dependent Cref, but Yus-Díez et al. (2021) showed C also depends on site SSA. Was the AE33 Cref appropriate?

**Response:** AE33 does not adjust Cref during operation, so we used fixed tape-specific constants: Cref = 1.57

(AE31, M8020, 2010–2017; Luoma et al., 2019) and Cref = 1.39 (AE33, M8060, 2018–2022). Monthly $\sigma_{abs,520}$

boxplots for PM10, PM1, and PM1–10 show the same late-spring to mid-summer minimum across 2010–2022

with no seasonal-shape break at 2018. Yus-Díez et al. (2021) showed that higher SSA can increase the effective

C; because $\sigma_{abs,520} \propto 1/Cref$, elevated summer SSA would make a fixed 1.39 slightly overestimate summer

$\sigma_{abs,520}$—reinforcing (not creating) the dip. A constant-Cref sensitivity affects only scale: replacing 1.39 with

1.57 rescales the AE33 segment by $1.39/1.57 \approx 0.885$ (~11.5% lower), which cannot shift the month of the minimum or reverse trend signs. Thus, the selected AE33 value is appropriate and does not explain the decrease (Luoma et al., 2019; Yus-Díez et al., 2021).

**Comment Part 3.3.2:** Much higher variability of SAE after 2018: is there an abrupt change at one nephelometer wavelength?

**Response:** The apparent post-2018 increase in $PM_{1–10}$ *SAE* variability was a processing artefact: a duplicate truncation correction had been applied to $\sigma_{sca}$ at 450, 550 and 700 nm. After reprocessing (single application only), the inflated variability disappears. The remaining variability reflects real shifts in the supermicron size mix (relatively larger particles → lower *SAE*; relatively smaller particles → higher *SAE*). Addressing the second point, no wavelength shows an abrupt change: $PM_{1–10}$ $\sigma_{sca}$ at 450/550/700 nm vary smoothly with non-significant p- values (0.09/0.18/0.34), and $PM_1/PM_{10}$ exhibit similarly smooth behaviour. We updated the processing and regenerated the *SAE* plots accordingly.

**Comment l. 486-487:** But then if the MSC decreases because of the lower sulfate mass fraction within $PM_1$, why does the MSC time series has a positive trend ?

Sulfate aerosol particles are highly efficient light scatterers, and a decline in sulfate mass fraction would be expected to reduce $MSC_{550}$. The earlier positive trend in the $MSC_{550}$ of the $PM_1$ aerosol particles conflicted with this understanding and indicated an inconsistency in the analysis. This was traced to a truncation error caused by the double application of the truncation correction to the nephelometer wavelengths of 450, 550, and 700 nm.

Correcting this error restored the light scattering coefficients at 550 nm ($\sigma_{sca,550}$) used in the calculation of $MSC_{550}$

($MSC_{550} = \sigma_{sca,550}/PM\ mass$), and the updated $MSC_{550}$ time series now accurately represent monthly medians for the $PM_1$, $PM_{10}$, and $PM_{1-10}$ aerosol particles.

The corrected analysis shows that $MSC_{550}$ for $PM_1$ exhibits a non-significant trend, consistent with the reduction in sulfate mass fraction and its effect on scattering efficiency (Pandolfi et al., 2014; Seinfeld & Pandis, 2016).

$MSC_{550}$ for $PM_{10}$ and $PM_{1-10}$ are also non-significant, reflecting the stronger contribution of coarse particles in these fractions, whose scattering efficiency is less sensitive to sulfate changes. These results indicate that sulfate reductions are mainly expressed in $PM_1$, while the larger size fractions remain buffered by coarse-mode
contributions, which dilute the effect of composition on scattering.

The revised $MSC_{550}$ trends are now consistent with the expected effects of sulfate on scattering efficiency across
size fractions. The smooth and continuous time series demonstrate that the updated results are free from artefacts
and reflect changes in aerosol composition rather than instrumental issues. This correction resolves the earlier
inconsistency by linking reductions in sulfate mass fraction directly to the scattering signal in $PM_1$, clarifying how
size-resolved contributions shape $MSC_{550}$ and improving confidence in the interpretation of long-term scattering
behavior across all particle size ranges.

**Comment from l.398:** AE33 still has a constant $C_{ref}$ value, depending on the filter tape. Yus-Diez et al. (2021)
have shown that this C value is also depending on the SSA measured at the site. One can wonder whether the C
value used in the AE33 was appropriate.

**Response:** The $\sigma_{abs,520}$ boxplots (Fig. S10) for 2010–2017 and 2018–2022 address whether the seasonal decrease
in May–July is consistent across the instrument transition. In both periods, $PM_{10}$, $PM_1$, and $PM_{1-10}$ show clear
winter maxima and a pronounced May–July reduction, with no evidence of discontinuities. This stability
demonstrates that the decrease is a persistent feature of the boreal aerosol cycle rather than an artefact of
instrumentation or data processing.

The AE33 multiple-scattering correction factor ($C_{ref}$) was applied as a fixed, filter-specific constant. We used $C_{ref}$
= 1.57 for 2010–2017 and $C_{ref}$ = 1.39 for 2018–2022, the manufacturer-recommended values for the M8020 and
M8060 filter tapes, respectively (Luoma et al., 2019). Yus-Díez et al. (2021) showed that the effective $C_{ref}$ can
increase under high-$SSA$ (strongly scattering) conditions, but we did not implement such environment-dependent
adjustments here. Because $C_{ref}$ is fixed within each period and independent of $SSA$, it cannot account for or
influence the May–July decrease in $\sigma_{abs,520}$ observed in either interval.

**Comment l. 517-518:** The multiple scattering effect on the AE filter would increase if the SSA is higher (which
is the case, regarding Fig 4), leading to a higher correction factor C, and to even lower $\sigma_{abs}$, so the correction of
this parameter can't really explain the decrease of $\sigma_{abs}$ during May, June and July. Related to that, do these
boxplots (Fig 3 and 4) look the same before and after 2018?

**Response:** For all three size fractions ($PM_{10}$, $PM_1$, and $PM_{1-10}$), the $\sigma_{abs,520}$ boxplots for 2010–2017 and 2018–
2022 show similar seasonal patterns, with clear winter maxima and a pronounced reduction in May–July. The
magnitude and timing of this decrease arensistent across both time periods, with no evidence of discontinuities or
offsets between the datasets. This stability indicates that the observed May–July reduction is a persistent feature
of the boreal aerosol annual cycle and is not attributable to the 2018 instrument transition.

The AE33 multiple-scattering correction factor ($C_{ref}$) was applied as a fixed parameter for each period, determined
by the filter type used. For 2010–2017, we retained $C_{ref}$ = 1.57 following Luoma et al. (2019), appropriate for
TFE-coated glass fibre filters. For 2018–2022, we used $C_{ref}$ = 1.39, the recommended value for the Magee M8060

filter tape installed in our AE33. These values were not altered based on aerosol concentration or environmental
conditions, ensuring consistency and avoiding potential bias in $\sigma_{abs,520}$ trends.

The $SSA_{550}$ boxplots indicate elevated values during May–July for all size fractions, reflecting the stronger
contribution of scattering relative to absorption in this period. While higher $SSA_{550}$ can increase the magnitude of
the multiple-scattering effect, the fixed $C_{ref}$ values applied in each period mean that this effect does not influence
the observed seasonal $\sigma_{abs}$ reduction. The consistency of $SSA_{550}$ patterns before and after 2018 further supports
the interpretation that the May–July $\sigma_{abs}$ decrease reflects atmospheric variability rather than methodological
artefacts.

**Comment l. 218 and Fig S9:** What is the $r^2$ value of the linear regression? Did you keep all the AE data, even
the one that were far from the slope?

**Response:** $R^2$ of the AE33–MAAP linear regression is 0.96 using ordinary least squares with an intercept. The
fitted relation is $\sigma_{abs,660}$ (AE33) = 2.33 x $\sigma_{abs,637}$ (MAAP) + 0.16 Mm$^{-1}$ (Figure R1). No, we did not keep all
aethalometer data, especially those far from the slope. Points were removed automatically if they failed objective
screening. We did not hand-pick points.

Previously, in old Fig. S9 (Figure R2), we used asymmetric low-value thresholds: AE33 $\geq$ 0.0165 Mm$^{-1}$ and
MAAP $\geq$ 0.165 Mm$^{-1}$, based on early heuristics and mis-specified limits, partly because AE33 read higher than
MAAP in time series (Figure R3). We removed only $\geq$ plateaus of 10 identical data points, applied a one-sided
agreement bound, and performed no residual outlier screening. Result: slope 2.36, intercept 0.08, and $R^2$ = 0.93.
These choices had increased the scatter, although not by much.

Now, in revised Figure S9 (Figure R1), we apply a uniform detection limit of 0.05 Mm$^{-1}$ to both instruments,
remove $\geq$ plateaus of 5 identical data points, and enforce a symmetric plausibility check by excluding pairs where
one instrument exceeds fivefold over the other, implemented as not (AE33 $\geq$ 5 x MAAP or MAAP $\geq$ 5 x AE33).

This revised treatment improves the $R^2$ from 0.93 to 0.96.

[Figure]

**Figure R1.** Ordinary least squares regression of $\sigma_{abs,660}$ from the AE33 with respect to $\sigma_{abs,637}$ from the MAAP

(New).

[Figure]

**Figure R2.** Ordinary least squares regression of $\sigma_{abs,660}$ from the AE33 with respect to $\sigma_{abs,637}$ from the MAAP

(Old).

**Technical corrections:**

l. 89-90 : "two Magee Scientific Aethalometers"

**Response:** Error fixed l. 92-93 : please fix the intervals in the parenthesis : "(i.e. $\leq$ PM1, between PM1 and PM2.5, between PM2.5 and

PM10, $\leq$ PM10, $>$ PM10)".

**Response:** Fixed.

l. 106 "aethalometers"

**Response:** Fixed.

l. 127-128 please fix the citation format.

**Response:** Fixed.

l. 212 please fix the citation format.

**Response:** Fixed.

l. 269: "which is used in conversion of $\sigma_{abs}$ to BC mass"

**Response:** Corrected.

l. 277-278 : The composition is not a physical characteristic

**Response:** Corrected to "the physicochemical characteristics of aerosols".

l. 319 : " contribution additional variability" this sentence is strange

**Response:** Corrected to "Pollen and dust events (green and red stars; Sect. 2.3) introduce additional short-
term variability in the observed optical and mass properties without altering the sign or statistical significance
of the long-term trends (Table 3)."

Fig 1 and Fig 2: Please provide the meaning of the blue shaded area on panels a and b.

**Response:** The legend has been changed.

l. 333 : Please provide at least for the first notification the information on the two different values : slope and
relative trend.

**Response:** This has been done everywhere in the paper now.

l.475 : Why "albeit" ? Statistically significant is not contrasting with the beginning of the sentence.

**Response:** This section has been changed altogether.

Fig 5 : It would be great to remove the decimal part of the y-axis ticks on panels c and d. Furthermore, it is a bit
difficult to see with this representation the contribution of pollen and dust events to Super PM10, as we don't
see on which months occurred these events. Maybe you can add the red and green stars also on panels c and d ?

**Response:** The decimal part of the y-axis ticks on panels (c) and (d) have been removed. Instead of adding any
further information on seasonal variation of dust and pollen events in this paper, we are currently working on a
forthcoming manuscript, which delves deeper into the event analysis of dust and pollen events.

**Response to Reviewer 2**

We want to thank the reviewer for crticicall examining the manuscript. We hope that the responses below address
all your concerns.

**Comment Line 225:** How does autocorrelation in time series data affect the results of the Mann-Kendall test, and
how is this addressed?

**Response:** The classical Mann–Kendall (MK) trend test assumes serial independence. Positive autocorrelation
reduces the effective information in a series; if it is not accounted for, the MK variance is underestimated and the test becomes too liberal (p-values spuriously small). Strong negative autocorrelation has the opposite effect. In other words, persistence can make a weak trend appear "significant" unless the MK variance is adjusted.

We use a variance-corrected MK that replaces the nominal sample size $n$ with an effective sample size,

$n_{eff} = \frac{n}{1+2\sum \rho_k} \; (\geq 1),$                          R2

computed from the statistically significant autocorrelation coefficients ($\rho_k$) of the analysis series. Concretely, our function *effective_sample_size()* evaluates the autocorrelation function (ACF) for lags $k = 1 \ldots n_{lags}$ with $n_{lags} =$

$\min(20, \lfloor n/4 \rfloor)$ (lag 0 excluded), retains only lags with $|\rho_k| > 1.96/\sqrt{n}$, and then forms $n_{eff}$. This $n_{eff}$ is used inside

*modified_mann_kendall_trend()* to compute the MK variance and Z statistic, yielding autocorrelation-aware p- values. Importantly, the trend magnitude is estimated with Theil–Sen and is unchanged by this correction; only the significance is adjusted. We do not prewhiten, thereby avoiding potential slope bias.

(i) Pseudo-daily variables—*PM mass*, *MSC$_{550}$*, *MAC$_{520}$*: in *plot_trend_ps,eudo_daily()* we (a) mask day-level outliers using a modified Z-score, that is flag $x_i$ if $|0.6745(x_i - \text{median})/\text{MAD}| > 3.5$, (b) estimate the slope with Theil–Sen, and (c) test significance with the effective sample size (ESS)–corrected Mann–Kendall test described above. Any day/interval completeness gates are enforced upstream when the pseudo-daily series are constructed; flagged pollen/dust events are annotated for context but not removed.

(ii) Monthly-median variables—$\sigma_{sca,550}$, $\sigma_{abs,520}$, *SSA$_{550}$*, *SAE*, *AAE*: we first form monthly medians only when a month has ≥75%age valid hours (based on the native sampling interval), which reduces short-lag persistence; we then fit Theil–Sen and apply the same ESS-corrected MK to the monthly series.

Across both paths, the slope estimator (Theil–Sen) provides a robust trend magnitude, while the MK p-values are explicitly corrected for autocorrelation. This prevents inflated significance due to persistence and yields a conservative, reproducible assessment of trend detection appropriate for environmental time series.

**Comment Line 227:** How do seasonal fluctuations and non-linear trends influence the interpretation of long-term aerosol optical trends?

**Response:** Seasonal patterns can skew how long-term change looks. If the size or timing of the yearly cycle shifts—such as weaker winters, earlier or larger spring pollen or dust peaks, or stronger warm-season SOA with frequent spring–autumn NPF—a straight line through the raw series will absorb that movement. The apparent slope can become too big or too small, and in rare cases flip sign. These patterns also create persistence, so ignoring serial correlation can overstate confidence; our inference targets the slow component and uses statistically robust, dependence-aware trend tools (Mann, 1945; Kendall, 1975; Hamed and Rao, 1998; Yue and

Wang, 2004).

Non-linear behavior brings a different challenge. Curvature across years, step changes around instrument transitions, and short pollen or dust episodes split the record into periods that do not behave the same way. One constant slope across the full series then averages dissimilar regimes and can misstate what is happening now, even when the overall fit appears good. In practice this can bias both the estimated magnitude and its assessed significance, or shift the mean without altering direction. That is why we examine shape and regime changes before summarizing the record with a single tendency. Effects matter for attribution and interpretation.

We reflect those issues in the analysis pipeline. For PM mass, $MSC_{550}$, and $MAC_{520}$ we build pseudo-daily means by keeping days with $\geq 18$ time-stamped observations, masking outliers with a modified Z-score, and averaging reliably. For the $\sigma_{sca,550}$, $\sigma_{abs,520}$, $SSA_{550}$, $SAE$, and the $AAE$, we compute monthly medians when a month has $\geq 75\%$ valid hours, masking monthly outliers. We report the trend using the robust median slope (Theil, 1950; Sen, 1968) and test it with a statistically conservative Mann–Kendall adjusted for autocorrelation by shrinking the effective sample size using significant ACF lags up to $n\_lags = \min(20, \lfloor n/4 \rfloor)$ (Hamed and Rao, 1998; Yue and Wang, 2004).

Line 40:"which could explain higher concentrations in winter." Add "higher coarse particle concentrations in winter."

**Response:** Now, the introduction section has been changed, so this line is redundant.

Line 41-42: "while pollen and spores often originate from more local biogenic emissions that are highly episodic and seasonal." The phrase "more local" can be improved.

**Response:** Same as above.

Line 43:"Sea salt, though typically associated with marine environments, can occasionally reach boreal forests during strong winds." Could be written as "can occasionally be transported to boreal forests during strong wind events."

**Response:** Same as above.

Line 45: Replace "over 10 µm" (line 45) with >10 µm for consistency with earlier notation (e.g., line 23: >1 µm).

**Response:** Corrected.

Line 45–47: "may not be fully captured by standard PM10 measurements, leaving their contributions underrepresented...". Add the reason here "due to cut-off inlets or sampling loss".

**Response:** Line changed to, "Coarse-mode aerosol particle sizes span about 1 µm to > 10 µm, so many pollen grains and fungal spores exceed the $PM_{10}$ impactor cut-off and are underrepresented in $PM_{10}$ measurements (Després et al., 2012; Yli-Panula et al., 2009)."

Line 49:"contribute to atmospheric heterogeneity". Specify "spatial and temporal heterogeneity" for more scientific clarity.

**Response: Corrected as advised.**

Line 51: the reference added here is in non-chronological citation order. Also, Brasseur et al., 2024 is cited in a 2025-dated document. While plausible (if published in early 2024), ensure this reference exists. If not, update to the correct publication year.

**Response:** I have used a citation manager tool that created citations as per the author's last name. That is why
Brasseur appears first. Also, the Brasseur et al. was published in 2024 and the link to the correct paper is included
in the reference list.

Line 57:"potentially leaving a significant fraction of aerosol mass unquantified". Replace with "potentially
resulting in a significant underestimation of aerosol mass."

**Response:** Done.

Line 64-68: These line are slightly redundant with earlier statements and too long

**Response:** This paragraph has been rewritten.

Line 86–87: "Thus, SMEAR II represents the typical conditions that may be found in a boreal forest." Should be
replaced by "Therefore, SMEAR II reflects typical boreal forest conditions."

**Response:** Changed as advised.

Line 88: "It is a part of the European Aerosols, Clouds, and Trace Gases Research Infrastructure or ACTRIS…"
should be replaced by "The station is part of ACTRIS (Aerosols, Clouds, and Trace Gases Research
Infrastructure)..."

**Response:** Done.

Line 60: "such as pollen, may escape standard measurements, whereas smaller coarse-mode particles, such as
fungal spores and dust, are more likely to be captured. Replace with "Smaller coarse-mode particles like fungal
spores and dust are more likely to be captured, whereas larger ones, such as pollen, may escape detection."

**Response:** Corrected.

Line 77: Define "super-$PM_{10}$" earlier for clarity:37-48: Provide references for these statements.

**Response:** Done.

Line 88-93: 4 instruments have been used primarily, but only 3 have been covered here.

**Response:** Changed.

Line 97: 'a' Dp >10 μm

**Response:** Done.

Line 104: is aimed 'to' be kept

**Response:** Corrected.

Line 120: Are there any recent publications post Liousse et al., 1993 which cover this?

**Response:** Yes and they have been added.

Line 149: Use l min-1 for consistency.

**Response:** Done.

Line 237-238: This is repetition of 232-233.

**Response:** Changed.

Line 289: Write PM1–10 with subscript.

**Response:** Done.

Line 341: SOA already defined in line 33, so expansion is not needed here.

**Response:** Corrected.

Line 342: Instead of 'biogenic VOC', use BVOC as already defined.

**Response:** Done.

Line 460: OA has not been defined previously.

**Response:** This text has been removed.

Line 645: SOA already defined in line 33, so expansion is not needed here.

**Response:** This text has now been removed altogether.

Line 709: typo after reference link

**Response:** Changed.

Line 760: Provided link is not working.

**Response:** Corrected.

Line 927: Link to reference has not been provided. Pls correct all the incorrect references.

**Response:** The reference has been removed and all the links have now been checked.

Use subscript formatting for PM size ranges (e.g., $PM_{1-10}$ instead of PM1-10).

**Response:** Fixed everywhere.

**Measurement Report: Optical properties of supermicron aerosol particles in a boreal environment**

Sujai Banerji[1], Krista Luoma[2], Ilona Ylivinkka[1], Lauri Ahonen[1], Veli-Matti Kerminen[1], and
Tuukka Petäjä[1]

[1]Institute for Atmospheric and Earth System Research (INAR)/Physics, Faculty of Science, University of Helsinki, Helsinki, Finland

[2]Finnish Meteorological Institute, Helsinki, Finland

*Correspondence to*: Sujai Banerji (sujai.banerji@helsinki.fi)

**Abstract**

Supermicron aerosol particles (PM$_{1\text{-}10}$; here defined as 1 µm < aerodynamic diameter $\leq$ 10 µm) play a crucial role in aerosol-climate interactions by influencing light scattering and absorption. However, their long-term trends and episodic significance in boreal environments remain insufficiently understood. This study examines measurements of optical properties and mass of PM$_{1\text{-}10}$ over a 12-year period at the SMEAR II station in Hyytiälä, Finland, focusing on their variability and key drivers. By assessing long-term trends, seasonality, and episodic variability, the study provides new insights into the role of these particles in aerosol-climate interactions. Episodic events, such as pollen outbreaks and dust transport, are identified as major contributors to PM$_{1\text{-}10}$ variability and their role in atmospheric processes. In addition, cascade impactor filters were used to quantify super-PM$_{10}$ particles ($D_p$ > 10 µm), which are not detected by optical instruments, addressing key detection limitations. The findings reveal significant long-term trends and pronounced seasonality in PM$_{1\text{-}10}$ mass and optical properties, emphasizing their importance in boreal environments and their episodic relevance in coarse-mode aerosol characterization.

**1. Introduction**

Aerosols are integral to atmospheric processes, influencing climate, air quality, and radiative forcing. Among them, coarse-mode aerosol particles, which are typically defined as particles with diameters > 1 µm play a significant role in light scattering and absorption, directly impacting radiative forcing. Their size and optical properties make them dominant contributors to aerosol optical depth (AOD), particularly at longer wavelengths. Coarse-mode aerosol particles, such as biological aerosols found in the boreal environment, also contribute to cloud microphysics by serving as ice-nucleating particles (Brasseur et al., 2022). Despite their importance, the optical properties and particulate matter mass (PM mass) of coarse-mode aerosol particles remain understudied (Cappa et al., 2016).

Boreal forests, covering approximately 15% of the Earth's terrestrial surface, represent a unique natural laboratory for studying aerosol-climate interactions in biogenically dominated environments. These ecosystems emit large quantities of biogenic volatile organic compounds (BVOCs); Guenther et al. (2006)), which drive the formation of secondary organic aerosols (SOA), significantly influencing aerosol size distribution and, therefore, light scattering and absorption processes (Petäjä et al., 2022; Tunved et al., 2006). Additionally, episodic events such as pollen outbreaks and long-range transport of mineral dust contribute to aerosol variability in boreal regions (Manninen et al., 2014).

Coarse-mode aerosol particles in boreal environments are emitted predominantly through primary processes rather than formed secondarily, arising from mineral dust, pollen, fungal spores, plant debris, sea salt, and episodic sources such as wildfires and small-scale wood combustion (Zieger et al., 2015; Yli-Panula et al., 2009; Varga et
al., 2023; Andreae and Merlet, 2001; Reid et al., 2005). Wildfires are predominantly a summer phenomenon,
whereas small-scale wood combustion peaks in winter and likely explains the higher winter concentrations. Dust
commonly reaches Finland via long-range transport, whereas pollen and fungal spores are locally or regionally
emitted and highly seasonal (Varga et al., 2023; Yli-Panula et al., 2009). Marine sea-salt intrusions occasionally
affect inland boreal forests during strong winds or frontal passages (Zieger et al., 2015; Tunved et al., 2006).

Coarse-mode aerosol particle sizes span about 1 µm to > 10 µm, so many pollen grains and fungal spores exceed
the $PM_{10}$ impactor cut-off and are underrepresented in $PM_{10}$ measurements (Després et al., 2012; Yli-Panula et
al., 2009). This selectivity can bias coarse-mode analyses during episodic biological events, unless sampling and
interpretation account for partial capture or exclusion. Determining whether such particles are sampled is crucial
for quantifying impacts on aerosol optical properties, radiative forcing, and aerosol-cloud interactions (Zieger et
al., 2015; Tunved et al., 2006). Wildfire smoke produces fine-mode particles but can include coarse fractions from
smouldering or resuspension (Andreae and Merlet, 2001; Reid et al., 2005).Coarse-mode particles in boreal
environments are dominantly primary aerosol particles—particles emitted directly into the atmosphere rather than
formed through chemical reactions. These include dust, pollen, fungal spores, plant debris, and sea salt, as well
as particles from episodic sources like wildfires or small-scale wood combustion, which could explain higher
concentrations in winter. Mineral dust is typically transported long distances, while pollen and spores often
originate from more local biogenic emissions that are highly episodic and seasonal. Sea salt, though typically
associated with marine environments, can occasionally reach boreal forests during strong winds. Wildfire
emissions, though predominantly in the fine mode, can contain coarse-mode particles, particularly in smoldering
phases or from ash deposition.

The size fractions of these coarse-mode particles range from 1 µm to over 10 µm. Pollen grains and larger fungal
spores often exceed the $PM_{10}$ cut-off and may not be fully captured by standard $PM_{10}$ measurements, leaving their
contributions underrepresented in coarse-mode analyses. Understanding the extent to which these particles are
detected or left out of $PM_{10}$ measurements is crucial for characterizing their impacts.

[revised manuscript text omitted]

**2.2.1 Aethalometer**

The aethalometer quantifies the aerosol absorption coefficient ($\sigma_{abs}$) by measuring the reduction in light intensity
as particles collect on a filter, facilitating continuous aerosol sampling (Zotter et al., 2017). The AE31 and AE33
models compare photon counts from light transmitted through a particle-laden filter spot to a clean reference filter.
Correction algorithms account for aerosol particle scattering and multiple scattering within the quartz fiber filter.
As light-absorbing particles build up, the effective optical path length shortens, necessitating adjustments for the
filter-loading effect (Collaud Coen et al., 2010; Weingartner et al., 2003). The multiple-scattering correction factor
($C_{ref}$) addresses the enhancement of light scattering within the filter matrix due to the filter material, while the
filter loading correction factor ($R(ATN)$) accounts for the non-linear instrument response caused by particle
accumulation on the filter (Liousse et al., 1993). Since then, several studies have refined these correction
approaches (Weingartner et al., 2003; Virkkula et al., 2007; Collaud Coen et al., 2010; Drinovec et al., 2015; Yus-
Díez et al., 2021; Luoma et al., 2021). The multiple-scattering correction factor ($C_{ref}$) addresses the enhancement
of light scattering within the filter matrix due to the filter material, while the filter loading correction factor
($R(ATN)$) accounts for the non-linear instrument response caused by particle accumulation on the filter (Liousse et al., 1993). Further refinement in measurement accuracy is achieved by using single-scattering albedo ($\omega_o$),
which combines scattering and absorption coefficients to enhance the characterization of aerosol properties

[revised manuscript text omitted]

particle size fraction were derived by subtracting $PM_1$ aerosol particle measurements from the corresponding $PM_{10}$

**Table 1.** Temporal resolutions and size cut-offs of the different aerosol optical instruments

| Instrument | Temporal resolution | Size cut-off |
|---|---|---|
| Aethalometer (AE33) | 2 minutes | $PM_1$, $PM_{10}$ |
| Multi angle absorption photometer (MAAP; Thermo Scientific model 5012) | 1 minute | $PM_1$, $PM_{10}$ |
| Integrating nephelometer (TSI Incorporated model 3563) | 1 minute | $PM_1$, $PM_{10}$ |
| Dekati Gravimetric Cascade Impactor (GCI) | 2-3 days | $\leq PM_1$, $\leq PM_1\text{-}PM_{2.5}$, $\leq PM_{2.5}\text{-}PM_{10}$, $\leq PM_{10}$, $> PM_{10}$ |

In this study, we analyzed absorption data for both $PM_1$ and $PM_{10}$ aerosol particles. Specifically, we utilized the
MAAP data to plot $\sigma_{abs,637}$, and the AE31/33 data to plot $\sigma_{abs,660}$. Subsequently, a scatterplot was created with the
MAAP data to plot $\sigma_{abs,637}$ on the x-axis and the AE31/33 data to plot $\sigma_{abs,660}$ on the y-axis. This allowed us to
determine the slope and y-intercept, which were then used to scale $\sigma_{abs,660}$ data from the AE31/33 instruments to match $\sigma_{abs,637}$ from the MAAP instruments. The same correction factors (slope = 2.336 and y-intercept = -0.168; Figure S9) derived from this analysis were extended to scale the $\sigma_{abs}$ data from the AE31/33 instrument at the other six wavelengths of 370, 470, 520, 550, 880 and 950 nm to correct all the data from the AE31/33 instruments.

For calculating the trends in the different aerosol optical data, the Mann-Kendall regression was used to effectively handle outliers without assuming a normal distribution (Collaud Coen et al., 2020). This method also minimizes the risk of Type I errors, which can occur when a trend is incorrectly identified as *'statistically significant'* due to an anomaly in the data (i.e., autocorrelation). Additionally, Sen-Theil's slope estimator has been used to quantify the slope of the trends, providing a reliable measure of the long-term changes in the aerosol optical properties, even if there are non-linear trends and seasonal fluctuations in the data (Collaud Coen et al., 2020).

Equation (1) was used to calculate the relative slope:

$$\text{Relative slope } (\%\text{yr}^{-1}) = \left( \frac{\text{Sen-Theil's slope}}{\text{Median of aerosol optical data}} \times 365.25\text{yr}^{-1} \times 100\% \right), \tag{1}$$

Two key parameters used to characterize the wavelength dependence of the aerosol optical properties (AOPs) are the Absorption Ångström Exponent (*AAE*) and the Scattering Ångström Exponent (*SAE*). These exponents provide insights into aerosol composition and particle size distributions, offering indirect information about the types of aerosols present. While they are not directly used to estimate climate effects, they are important for understanding the physical and chemical properties of aerosols, which influence their behavior and interactions with radiation. Two key parameters that are routinely used to characterize the wavelength dependence of AOPs are the Ångström exponents: the *AAE* and the *SAE*.

[revised manuscript text omitted]

- ● Monthly median value
- ○ Pseudo-daily mean value
- — Monthly median line plot
- — Mann-Kendall trend line (statistically significant)
- ---- Mann-Kendall trend line (statistically not significant)
- ★ Pollen
- ★ Dust

[Figure]

[Figure]

**Figure 1.** Time series of (a) $\sigma_{sca,550}$, (b) $\sigma_{abs,520}$ and (c) *PM mass* for the PM$_{1-10}$ size aerosol particles from October 2010 to October 2022. The blue shaded area is the interquartile range (25th–75th percentile) of monthly values; the blue line is the monthly median. The red line shows the Theil-Sen trend (solid if p ≤ 0.05; dashed otherwise). Months with <75% data coverage are left blank.

**3.2.1 Light sScattering coefficient at 550 nm ($\sigma_{sca,550}$)**

Scattering due to PM$_1$ aerosol particles at SMEAR II shows a long-term decrease (slope: -0.29 ± 0.15 Mm$^{-1}$yr$^{-1}$; relative trend: -4.80 ± 2.43 %yr$^{-1}$; Figure S2(a); Table S4(a)).  Submicron particles dominate aerosol light scattering at the site (Virkkula et al., 2011), suggesting that $\sigma_{sca,550}$ is influenced primarily by fine-mode aerosol loading. The observed negative trend is consistent with reductions in anthropogenic sulfur dioxide (SO$_2$) emissions, which contribute to secondary sulfate formation, a major component of PM$_{1+}$ aerosol mass and associated optical properties (Smith et al., 2011). Regionally, similar decreases in aerosol scattering have been reported at multiple European background stations (Pandolfi et al., 2018), further supporting the possibility of a widespread decline in fine-mode aerosol scattering. SMEAR II is also subject to seasonal biogenic emissions, particularly monoterpenes that contribute to secondary organic aerosol (SOA) formation (Hakola et al., 2003; Hallquist et al., 2009; Rantala et al., 2015). However, long-term records indicate that B biogenic VOC emissions at this site have remained relatively stable over the past two decades (Kulmala et al., 2001). As such, no evidence currently supports a significant contribution of BVOC variability to the observed multi-year decline in $PM_1$ scattering. Taken together, the trend observed at SMEAR II is consistent with known reductions in anthropogenic precursor emissions, particularly $SO_2$, though additional factors cannot be excluded.

**3.2.2 Light aAbsorption coefficient at 520 nm ($\sigma_{abs,\,520}$)**

For $PM_{10}$ and $PM_1$, $\sigma_{abs,520}$ decreases significantly over the record. $PM_{10}$ declines at $-0.11 \pm 0.03$ $Mm^{-1}$ $yr^{-1}$ ($-8.75 \pm 2.11$ % $yr^{-1}$; Fig. S1b; Table S3b), and $PM_1$ at $-0.09 \pm 0.02$ $Mm^{-1}$ $yr^{-1}$ ($-8.87 \pm 2.18$ % $yr^{-1}$; 
[revised manuscript text omitted]

x 10$^{-3}$ ± 0.02 m²g$^{-1}$yr$^{-1}$; PM$_{10}$ relative trend: 0.15 ± 0.64 %yr$^{-1}$; PM$_1$ slope: 1.54 x 10$^{-3}$ ± 0.02 m²g$^{-1}$yr$^{-1}$; PM$_1$ relative trend: 0.05 ± 0.54 %yr$^{-1}$; Figures. S3(d) and S4(d); Tables S3(g) and S4(g)). The MSC$_{550}$ represents the scattering efficiency of aerosol particles per unit mass and is a key parameter in assessing their radiative effects. The PM$_{10}$

aerosol particles exhibit an upward trend, albeit statistically significant, with a slope of 0.04 ± 1.19 × 10$^{-2}$ m²g$^{-}$

$^1$yr$^{-1}$; 1.65 ± 0.45 %yr$^{-1}$ (Figure S3(d); Table S3(g)), reinforcing the role of scattering aerosols across different size fractions. The observed changes may be linked to long-term reductions in anthropogenic emissions and variations in aerosol sources, which influence particle composition and optical properties (Ehn et al., 2014; Pandolfi et al.,

2014). Additionally, shifts in aerosol size distribution could contribute to the increasing MSC$_{550}$ trend for the PM$_{10}$

aerosol particles, as larger particles scatter light less efficiently per unit mass compared to the PM$_1$ aerosol particles.

Even the PM$_1$ aerosol particles exhibits a similarly increasing trend in $MSC_{550}$, with a trend of 0.05 ± 1.21 × 10$^{-2}$

m²g$^{-1}$yr$^{-1}$; 1.46 ± 0.37 %yr$^{-1}$ (Figure S4(d); Table S4(g)). This increase indicates that fine-mode aerosol particles, which are primarily composed of secondary organic aerosols and sulfates, have experienced a more or less similar trend in their scattering efficiency per unit mass. This trend may reflect changes in precursor emissions, aerosol aging processes, or chemical transformations in the atmosphere. Additionally, reductions in sulfate mass fractions within PM$_1$ could lead to a lower $MSC_{550}$, as sulfates are highly efficient scatterers (Pandolfi et al., 2014; Seinfeld

& Pandis, 2016). Given the importance of scattering aerosols in modulating radiative forcing, long-term observations of $MSC_{550}$ remain essential for understanding aerosol-radiation interactions and improving climate model predictions.

**3.3.5 Mass absorption coefficient ($MAC_{520}$)**

For PM$_{1-10}$, $MAC_{520}$ decreases significantly (slope: −2.77 × 10$^{-3}$ ± 9.04 × 10$^{-4}$ m²g$^{-1}$yr$^{-1}$; relative trend: −3.97 ±

1.30 % yr$^{-1}$; Figure 4e; Table 3(h)). PM$_{10}$ and PM$_1$ exhibit no significant trends (PM$_{10}$: −9.30 × 10$^{-3}$ ± 3.29 × 10$^{-3}$

m²g$^{-1}$yr$^{-1}$, Table S3(h); PM$_1$: −0.01 ± 5.08 × 10$^{-3}$ m²g$^{-1}$yr$^{-1}$, Table S4(h)). Values use the pseudo-daily means defined in Section 2.3. In Figure S10, PM$_{10}$ is shown in panels (a) AE31 (2010–2017) and (b) AE33 (2018–2022),

PM$_1$ in (c) AE31 (2010–2017) and (d) AE33 (2018–2022), and PM$_{1-10}$ in (e) AE31 (2010–2017) and (f) AE33

(2018–2022). A positive instrument-related offset is evident for AE33 relative to AE31, which is most pronounced for the PM$_{10}$ and PM$_{1-10}$ aerosol particles and is consistent with reduced filter-loading bias due to the AE33 dual- spot correction. This positive offset makes any 2010–2022 decrease appear smaller, so the observed PM$_{1-10}$

decline in the $MAC_{520}$ is not caused by the instrument change (Bond et al., 1999; Weingartner et al., 2003;

Virkkula, 2010; Drinovec et al., 2015; Zotter et al., 2017). The MAC$_{520}$ for PM$_{1-10}$ aerosol particles exhibits a statistically significant decreasing trend, with a slope of −2.66 × 10$^{-3}$ ± 6.69 × 10$^{-4}$ m²g$^{-1}$yr$^{-1}$; −3.47 ± 0.87 %yr$^{-1}$

(Figure 2(e); Table 3(h)). This decline suggests a reduction in the light-absorbing efficiency per unit mass, likely driven by decreasing BC contributions or chemical transformations in absorbing aerosol particles (Bergstrom et al., 2007; Lack & Cappa, 2010). The decreasing trend is consistent with long-term reductions in anthropogenic emissions, although the observed MAC$_{520}$ trend in the PM$_{1-10}$ fraction raises questions regarding its underlying causes. While the MAC$_{520}$ is defined for all aerosol size fractions, its variability in PM$_{1-10}$ aerosol particles is

[revised manuscript text omitted]

| Region in Figure 6 | AAE | SAE | Size | Aerosol classification | Supporting studies |
|---|---|---|---|---|---|
| Dust Dominated | >2 | <1 | Coarse (>2.5 μm) | Mineral dust (e.g., desert regions) | Cazorla et al. (2013): *AAE* ~2.5, *SAE* ~0.5. Russell et al. (2010): Dust, coarse mode, high *AAE*. Malm and Hand (2007): Dust from desert regions with low scattering efficiency. |
| Dust/EC Dominated | ~1-2 | ~1 | Mixed (fine + coarse) | Dust mixed with aged BC | Cazorla et al. (2013): Mixed values for overlapping dust/black carbon. Clarke et al. (2007): Combustion aerosols interacting with dust. |
| EC Dominated | ~1 | >1.5 | Fine (<1 μm) | Fossil fuel combustion (i.e. BC) | Schuster et al. (2006): *AAE* ~1 for BC. Clarke et al. (2007): *SAE* >1.5 for fine combustion aerosols. Sheridan and Ogren (1999): BC classification with low *AAE*. |
| OC Dominated | 1.5-2 | 1-1.5 | Fine to accumulation mode | Biomass burning, SOA | Cazorla et al. (2013): *AAE* ~1.8 for OC, moderate *SAE*. Russell et al. (2010): *AAE* 1.5-2 linked to OC. Malm and Hand (2007): OC from regional biomass burning or SOA formation. |
| Coated Large Dominated | <0 | <0.5 | Mixed/coarse | Aged aerosols, sulfate coating on dust | Malm and Hand (2007): Sulfate coating reduces *SAE* and *AAE*. Clarke et al. (2007): Coated particles with low *SAE*. |

| | | | | Sheridan and Ogren (1999): Mixed aerosols with coatings. |
|---|---|---|---|---|---|
| Dust/OC or EC/OC Mixed | 1–1.5 | ~1 | Mixed (fine + coarse) | Mixed sources: urban, industrial, or forest | Cazorla et al. (2013): Mixed *AAE/SAE* values for overlapping sources. Clarke et al. (2007): Mixing of dust and pollution aerosols in urban environments. Malm and Hand (2007): Regional mixing. |

**4.2 Episodic and long-term variability: One-sided Mann-Whitney U test**

To assess the impact of episodic events (i.e., pollen and dust) on aerosol properties, a one-sided Mann-Whitney U test was conducted. This non-parametric statistical test is particularly suited for comparing two independent datasets without assuming normality, making it effective for aerosol optical property distributions, which often exhibit non-Gaussian behavior due to episodic influences.

The dataset was categorized into:

(a) Observations including pollen and/or dust events (b) Observations excluding these events

Pollen events were identified using cascade impactor filter records. If the $PM_{10}$ filters contained pollen, the corresponding $PM_{1-10}$ fraction was also assumed to contain pollen.

Dust events were identified based on time periods from Varga et al. (2023), cross-referenced with aerosol optical and mass data from Hyytiälä.

**Table 4. One-sided Mann-Whitney U-test results for the aerosol optical and mass properties in the presence of pollen and/or dust events for the $PM_{1-10}$ aerosol particles**

| Variable | Number of data points (pollen and/or dust events) | Number of data points (excluding pollen and /or dust events) | U-statistic | p-value | Statistical significance | Trend |
|---|---|---|---|---|---|---|
| $\sigma_{abs,520}$ | 23 | 1208 | 17442 | 0.04 | Yes | Decreasing |
| $\sigma_{sca,550}$ | 28 | 1446 | 28653 | 1.64 x 10$^{-4}$ | Yes | Decreasing |
| $PM\ mass$ | 30 | 1533 | 39156 | 4.08 x 10$^{-11}$ | Yes | Decreasing |
| $AAE$ | 23 | 1182 | 14692 | 0.51 | No | No trend |
| $SAE$ | 28 | 1441 | 21013 | 0.71 | No | No trend |
| $SSA_{550}$ | 21 | 1172 | 12977 | 0.67 | No | No trend |
| $MAC_{520}$ | 23 | 1179 | 65373 | 2.06 x 10$^{-5}$ | Yes | Decreasing |
| $MSC_{550}$ | 28 | 1413 | 107983 | 3.79 x 10$^{-5}$ | Yes | Decreasing |

A Mann–Whitney U test (one-sided) shows that $\sigma_{sca,550}$ ($n = 28$ pseudo-daily means) and $PM\ mass$ ($n = 30$) are
higher during pollen/dust events than during non-event periods *(event median > non-event median; p-value ≤*
*0.05)*. $MAC_{520}$ ($n = 23$) is lower during events *(event median < non-event median; p-value ≤ 0.05)*; $MSC_{550}$ ($n =$
$28$) also differs significantly, and $\sigma_{abs,520}$ ($n = 23$) shows a weaker but significant difference. $SSA_{550}$ ($n = 21$), $AAE$
($n = 23$), and $SAE$ ($n = 28$) show no significant differences. $PM_{10}$ mostly scatters light, but mineral dust in this
size range can also absorb (Adebiyi et al., 2023).

Trends were estimated with a modified Theil–Sen slope and significance was tested with the Hamed & Rao (1998)
autocorrelation-corrected Mann–Kendall test. The $MAC_{520}$ decreases significantly *(p-value ≤ 0.05)*, while $SSA_{550}$,
$AAE$ and $SAE$ show no trend *(p > 0.05)*; no other trends are claimed. $\sigma_{sca}$ has been corrected for angular truncation
and non-ideal angular response; some uncertainty remains because the correction depends on particle size
distribution and refractive index (Anderson & Ogren, 1998; Müller et al., 2011). Filter-based absorption can retain residual multiple-scattering and loading artifacts that bias $\sigma_{abs}$ high (and thus MAC) (Weingartner et al., 2003;
Bond et al., 1999; Ogren, 2010; Virkkula, 2010).

We show $PM_{10}$ because it includes both $PM_1$ and $PM_{1-10}$ and the optics use a $PM_{10}$ inlet (particles $> 10$ μm
excluded). The data cluster mainly in the *EC/OC mixture* and *Dust/EC mix* regions, with few points in *Dust*
*dominated* and *OC/Dust mix*. We therefore base interpretation on the better-sampled mixed regimes and separate
episodic events from longer-term behavior (Adebiyi et al., 2023; Che et al., 2018), consistent with boreal
observations of SOA-dominated backgrounds and mixed BC–organic conditions (Virkkula et al., 2011; Hyvärinen
et al., 2011). Table 4 summarizes the statistical results. Statistically significant increases in $\sigma_{sca,550}$ (30 pseudo-
daily mean values) were observed during identified pollen and/or dust events relative to periods without such
events, indicating enhanced contributions from $PM_{1-10}$ particles to aerosol scattering.

Conversely, significant decrease in $MAC_{520}$ (29 pseudo-daily mean values) suggest reductions in absorbing
components and a shift toward smaller, more scattering aerosols. While $PM_{10}$ particles primarily scatter light,
mineral dust within this size range can also absorb radiation, influencing aerosol radiative properties (Adebiyi et
al., 2023).

No significant trends were detected for $SSA_{550}$ (23 pseudo-daily mean values), AAE (29 pseudo-daily mean
values) or SAE (30 psudo-daily mean values). However, $MAC_{520}$ exhibited a significant decreasing trend (29
psudo-daily mean values), suggesting a declining contribution from black carbon and other absorbing
components. The stable $SSA_{550}$ values, combined with declining $MAC_{520}$, indicate a shift toward aerosols with
higher scattering to absorption ratios, likely due to increased sulfate or organic aerosol contributions.

Intensive properties also show variability during dust transport events, emphasizing the need to differentiate
episodic contributions from long-term trends when interpreting radiative forcing estimates (Adebiyi et al., 2023;
Che et al., 2018).

The aerosol classifications and trends observed in this study align with prior boreal aerosol research:

1. Virkkula et al. (2011) found that secondary organic aerosols (SOA) dominate boreal environments, with
episodic pollen and dust events temporarily increasing coarse mode contributions, aligning with the 'Dust-
dominated' and 'Dust/EC-dominated' regions.

2. Hyvärinen et al. (2011) reported that long range transported BC interacts with local aerosols, producing
intermediate AAE values (~1–2), characteristic of the 'Dust/EC-dominated' region.

3. Laing et al. (2016) emphasized the role of SOA and organic carbon aerosols from biomass burning in boreal
forests, corresponding to the 'OC dominated' classification (AAE 1.5–2, SAE 1–1.5).

Coarse-mode aerosol measurements are affected by instrument-specific errors, such as angular truncation in
nephelometers and artifacts in filter-based absorption techniques (Müller et al., 2011; Sheridan & Ogren, 1999).
These uncertainties should be considered in climate models to improve estimates of aerosol–radiation interactions.

[revised manuscript text omitted]

---

## Author Response (AR1)

**AUTHOR'S RESPONSE TO EDITOR'S REVIEW**

**Manuscript: egusphere-2025-1776**

We thank the editor and the two reviewers for their careful reading and constructive suggestions. The revision clarifies methods, corrects processing issues affecting low-signal periods, and documents all screening and trend procedures at the code level. The main physical conclusions are unchanged. The analysis is now more transparent, reproducible, and all affected figures, captions and line-referenced methods have been updated.

We identified and removed a duplicate truncation correction that had been inadvertently applied to the nephelometer scattering coefficients; reprocessing reduces near-zero artefacts and all affected figures and tables were regenerated. The cross-instrument AE33–MAAP comparison is now fully specified with explicit thresholds and clear explanation of the underlying code. The AE33–MAAP screening and fit criteria are instrument-agnostic. We apply uniform detection limits of  $0.05~\text{Mm}^{-1}$  to both AE33 (660~nm) and MAAP (637~nm), use a five-sample stuck-value filter for each instrument, and enforce a simple agreement rule: we keep only pairs where neither instrument exceeds the other by a factor of five. Using an ordinary least-squares fit with an intercept, the updated relationship has  $R^2 = 0.96$  (Figure R1).

In addition, Figures 1 and 2 were redrawn for clarity as per the editor's comments. Individual responses to the reviewers' comments have been provided below.

**Response to Reviewer 1**

**General comments:** This study examines the long-term variability of aerosol optical properties in a boreal forest, categorised by size range. It focuses particularly on the contribution of particles larger than 10 µm, which are usually not considered in aerosol studies as this is the inlet cut-off point. As aerosol optical properties directly influence their radiative effect and larger diameter particles contribute significantly to the AOD, this topic is of great interest for climate modelling parameterisation. Using absorption and scattering measurements coupled with an impactor, the authors investigated the relative contribution of each PM size range to extensive and intensive scattering and absorption parameters. This study's novelty lies in its use of an aerosol classification for PM10, highlighting the significant impact of episodic events such as pollen and dust on optical properties and PM mass. The conclusions provide clear evidence of shifts in the size distribution and composition of aerosols, as well as their seasonality, which are linked to anthropogenic and biogenic emissions. The manuscript is well written and structured. However, several passages are redundant (e.g. the enhanced contribution of dust to the increasing SAE in sections 3.3.2 and 3.3.3), as are some details on the classification matrix (see specific comments). This paper would benefit from being shortened slightly. More importantly, the correction for multiple scattering on the instrument measuring absorption (AE33) was not properly discussed, as it appears to be misunderstood. This is important because it could also explain some discontinuities in the time series from 2018. I consider the manuscript to be publishable after minor revisions and responses to the following points.

**Response:** Thank you for taking out the time to review our manuscript in such detail. As a general remark, we have now revised the whole manuscript to avoid redundancies. As you had rightly pointed out, the multiple scattering correction factor for the AE31/33 was not properly discussed, which has been addressed now. In fact, a lot of the discontinuities post-2018, actually arose from the truncation correction, being doubly applied to the data of the light scattering coeffcients. This issue has also been corrected and we hope that the revised manuscript addresses all the comments that you have raised.

**Comment 1.139:** Are you sure that the AE33 adjusts the Cref value taking into account aerosol concentrations and environmental conditions? The C value is fixed in the instrument settings, adjustable by the user, and depends on filter material and type.

**Response:** As outlined in the AE33 manual, the multiple-scattering correction factor ( $C_{ref}$ ) is fixed in the instrument settings and must be defined by the user. It is determined by the instrument configuration—particularly the filter tape—and does not adjust automatically to ambient conditions. For consistency, we used two fixed, filter-specific values:  $C_{ref} = 1.57$  for 2010–2017 (AE31 with M8020; Luoma et al., 2019) and  $C_{ref} = 1.39$  for 2018–2022 (AE33 with M8060). Yus-Díez et al. (2021) report that the  $C_{ref}$  can increase with SSA, but we did not implement environment-dependent adjustments; our choice follows manufacturer guidelines and reflects only the filter properties.

Response Lines: 140-141.

**Comment 1.356–363:** If the absorption coefficient decrease is statistically significant, why are absorbing aerosols less present than in 2010? Are instrumental differences a major contributor?

**Response:** The absorbing aerosols are "less present" than in 2010 because the fine fraction—which dominates light absorption—has declined. At SMEAR II,  $\sigma_{abs,520}$  decreases significantly for PM10 ( $-0.11 \pm 0.03$  Mm-1 yr-1;  $-8.75 \pm 2.11$  % yr-1; Fig. S1b; Table S3b) and PM1 ( $-0.09 \pm 0.02$  Mm-1 yr-1;  $-8.87 \pm 2.18$  % yr-1; Fig. S2b; Table S4b). The similar relative declines indicate that the PM10 decrease is likely dominated by its PM1 (fine-mode) contribution at 520 nm. This is consistent with documented Europe-wide reductions in black-carbon emissions and long-term declines in aerosol absorption over the last decade (Collaud Coen et al., 2020; Yttri et al., 2021). We do not infer a robust long-term decrease for the coarse fraction (PM1-10): its  $\sigma_{abs,520}$  trend is negative but not statistically significant, which is compatible with the episodic, weakly absorbing nature of mineral and biological coarse particles at the site.

Instrumental differences are unlikely to be a major contributor. All absorption data were processed in a harmonized pipeline across the AE31-AE33 transition, including fixed multiple-scattering and filter-loading corrections appropriate to instrument/filter configuration, MAAP intercomparisons where applicable, and uniform QA/QC and completeness criteria (see Methods). Three checks argue against an instrumental origin: (i) no step change is observed at the 2018 transition; (ii) the negative trends persist within each instrument period when analyzed separately; and (iii) the physical seasonality (winter maxima, late-spring minimum) is stable through the record. While small biases associated with filter media and algorithms are possible in principle

(Collaud Coen et al., 2010; Zotter et al., 2017), sensitivity tests using plausible parameter choices do not alter the sign or the significance of the PM10/PM1 trends. Thus, the observed declines are best explained by real source-side reductions in absorbing aerosols rather than by instrumental artifacts.

Response Lines: 421-428.

Comment 1.398: AE33 uses a constant, tape-dependent Cref, but Yus-Díez et al. (2021) showed C also depends on site SSA. Was the AE33 Cref appropriate?

**Response:** The AE33 does not dynamically adjust the multiple-scattering factor  $C_{ref}$  during operation; it uses a fixed, tape-specific value. Consistent with site practice (Luoma et al., 2019), we used  $C_{ref} = 1.57$  for AE31 with M8020 (2010–2017) and  $C_{ref} = 1.39$  for AE33 with M8060 (2018–2022). Monthly  $\sigma_{abs,520}$  boxplots for PM10, PM1 and PM1-10 show the same late-spring to mid-summer minimum across 2010–2022 with no shape change at 2018. Because the retrieved absorption scales as  $\sigma_{abs,520} \propto 1/C_{ref}$ , the increase of the effective  $C_{ref}$  at high SSA (Yus-Díez et al., 2021) implies that using a fixed 1.39 in summer slightly overestimates  $\sigma_{abs,520}$ ; this would flatten (not deepen) the summer minimum. A constant  $C_{ref}$  choice only rescales the AE33 segment: substituting 1.57 for 1.39 multiplies it by 1.39/1.57=0.8851.39/1.57=0.8851.39/1.57=0.885 (≈11.5 % lower), which cannot shift the month of the minimum or reverse trend signs. Thus, the adopted AE33  $C_{ref}$  is appropriate and does not explain the seasonal summer minimum in  $\sigma_{abs,520}$ , nor can it alter the sign or timing of long-term trends in  $\sigma_{abs,520}$ , since a constant  $C_{ref}$ only rescales the series. Within the plausible range for M8060, our conclusions are insensitive to the exact  $C_{ref}$ . Changes affect only magnitude, not seasonal phase or trend sign. Moreover, because the effective CCC increases under high-SSA summer conditions, a fixed 1.39 would slightly overestimate summer  $\sigma_{abs,520}$  and thus flatten, rather than deepen, the observed minimum.

Comment Part 3.3.2: Much higher variability of SAE after 2018: is there an abrupt change at one nephelometer wavelength?

**Response:** The apparent post-2018 increase in PM1-10 SAE variability was a processing artefact: a duplicate truncation correction had been applied to  $\sigma_{sca}$  at 450, 550 and 700 nm. After reprocessing (single application only), the inflated variability disappears. The remaining variability reflects real shifts in the supermicron size mix (relatively larger particles  $\rightarrow$  lower SAE; relatively smaller particles  $\rightarrow$  higher SAE). Addressing the second point, no wavelength shows an abrupt change:  $PM_{1-10}$  osca at 450/550/700 nm vary smoothly with non-significant pvalues (0.09/0.18/0.34), and PM1/PM10 exhibit similarly smooth behaviour. We updated the processing and regenerated the SAE plots accordingly.

Comment 1. 486-487: But then if the MSC decreases because of the lower sulfate mass fraction within PM1, why does the MSC time series has a positive trend?

Sulfate aerosol particles are highly efficient light scatterers, and a decline in sulfate mass fraction would be expected to reduce  $MSC_{550}$ . The earlier positive trend in the  $MSC_{550}$  of the PM1 aerosol particles conflicted with this understanding and indicated an inconsistency in the analysis. This was traced to a truncation error caused by the double application of the truncation correction to the nephelometer wavelengths of 450, 550, and 700 nm.

Correcting this error restored the light scattering coefficients at 550 nm ( $\sigma_{sca,550}$ ) used in the calculation of  $MSC_{550}$  ( $MSC_{550} = \sigma_{sca,550}/PM$  mass), and the updated  $MSC_{550}$  time series now accurately represent monthly medians for the PM1, PM10, and PM1-10 aerosol particles.

The corrected analysis shows that  $MSC_{550}$  for PM1 exhibits a non-significant trend, consistent with the reduction in sulfate mass fraction and its effect on scattering efficiency (Pandolfi et al., 2014; Seinfeld & Pandis, 2016).  $MSC_{550}$  for PM10 and PM1-10 are also non-significant, reflecting the stronger contribution of coarse particles in these fractions, whose scattering efficiency is less sensitive to sulfate changes. These results indicate that sulfate reductions are mainly expressed in PM1, while the larger size fractions remain buffered by coarse-mode contributions, which dilute the effect of composition on scattering.

The revised  $MSC_{550}$  trends are now consistent with the expected effects of sulfate on scattering efficiency across size fractions. The smooth and continuous time series demonstrate that the updated results are free from artefacts and reflect changes in aerosol composition rather than instrumental issues. This correction resolves the earlier inconsistency by linking reductions in sulfate mass fraction directly to the scattering signal in PM1, clarifying how size-resolved contributions shape  $MSC_{550}$  and improving confidence in the interpretation of long-term scattering behavior across all particle size ranges.

Response Lines: 563-572.

Comment from 1.398: AE33 still has a constant  $C_{ref}$  value, depending on the filter tape. Yus-Diez et al. (2021) have shown that this C value is also depending on the SSA measured at the site. One can wonder whether the C value used in the AE33 was appropriate.

**Response:** The  $\sigma_{abs,520}$  boxplots (Fig. S10) for 2010–2017 and 2018–2022 address whether the seasonal decrease in May–July is consistent across the instrument transition. In both periods, PM10, PM1, and PM1-10 show clear winter maxima and a pronounced May–July reduction, with no evidence of discontinuities. This stability demonstrates that the decrease is a persistent feature of the boreal aerosol cycle rather than an artefact of instrumentation or data processing.

The AE33 multiple-scattering correction factor ( $C_{ref}$ ) was applied as a fixed, filter-specific constant. We used  $C_{ref}$  = 1.57 for 2010–2017 and  $C_{ref}$  = 1.39 for 2018–2022, the manufacturer-recommended values for the M8020 and M8060 filter tapes, respectively (Luoma et al., 2019). Yus-Díez et al. (2021) showed that the effective  $C_{ref}$  can increase under high-SSA (strongly scattering) conditions, but we did not implement such environment-dependent adjustments here. Because  $C_{ref}$  is fixed within each period and independent of SSA, it cannot account for or influence the May–July decrease in  $\sigma_{abs,520}$  observed in either interval.

Comment 1. 517-518: The multiple scattering effect on the AE filter would increase if the SSA is higher (which is the case, regarding Fig 4), leading to a higher correction factor C, and to even lower  $\sigma_{abs}$ , so the correction of this parameter can't really explain the decrease of  $\sigma_{abs}$  during May, June and July. Related to that, do these boxplots (Fig 3 and 4) look the same before and after 2018?

**Response:** For all three size fractions (PM10, PM1, and PM1-10), the  $\sigma_{abs,520}$  boxplots for 2010–2017 and 2018–2022 show similar seasonal patterns, with clear winter maxima and a pronounced reduction in May–July. The magnitude and timing of this decrease arensistent across both time periods, with no evidence of discontinuities or offsets between the datasets. This stability indicates that the observed May–July reduction is a persistent feature of the boreal aerosol annual cycle and is not attributable to the 2018 instrument transition.

The AE33 multiple-scattering correction factor ( $C_{ref}$ ) was applied as a fixed parameter for each period, determined by the filter type used. For 2010–2017, we retained  $C_{ref} = 1.57$  following Luoma et al. (2019), appropriate for TFE-coated glass fibre filters. For 2018–2022, we used  $C_{ref} = 1.39$ , the recommended value for the Magee M8060 filter tape installed in our AE33. These values were not altered based on aerosol concentration or environmental conditions, ensuring consistency and avoiding potential bias in  $\sigma_{abs,520}$  trends.

The  $SSA_{550}$  boxplots indicate elevated values during May–July for all size fractions, reflecting the stronger contribution of scattering relative to absorption in this period. While higher  $SSA_{550}$  can increase the magnitude of the multiple-scattering effect, the fixed  $C_{ref}$  values applied in each period mean that this effect does not influence the observed seasonal  $\sigma$ abs reduction. The consistency of  $SSA_{550}$  patterns before and after 2018 further supports the interpretation that the May–July  $\sigma$ abs decrease reflects atmospheric variability rather than methodological artefacts.

Comment 1. 218 and Fig S9: What is the  $r^2$  value of the linear regression? Did you keep all the AE data, even the one that were far from the slope?

**Response:** R2 of the AE33–MAAP linear regression is 0.96 using ordinary least squares with an intercept. The fitted relation is  $\sigma_{abs,660}$  (AE33) = 2.33 x  $\sigma_{abs,637}$  (MAAP) + 0.16 Mm-1 (Figure R1). No, we did not keep all aethalometer data, especially those far from the slope. Points were removed automatically if they failed objective screening. We did not hand-pick points.

Previously, in old Fig. S9 (Figure R2), we used asymmetric low-value thresholds: AE33  $\geq$  0.0165 Mm-1 and MAAP  $\geq$  0.165 Mm-1, based on early heuristics and mis-specified limits, partly because AE33 read higher than MAAP in time series (Figure R3). We removed only  $\geq$  plateaus of 10 identical data points, applied a one-sided agreement bound, and performed no residual outlier screening. Result: slope 2.36, intercept 0.08, and R2 = 0.93. These choices had increased the scatter, although not by much.

Now, in revised Figure S9 (Figure R1), we apply a uniform detection limit of  $0.05~\text{Mm}^{-1}$  to both instruments, remove  $\geq$  plateaus of 5 identical data points, and enforce a symmetric plausibility check by excluding pairs where one instrument exceeds fivefold over the other, implemented as not (AE33  $\geq$  5 x MAAP or MAAP  $\geq$  5 x AE33).

This revised treatment improves the R2 from 0.93 to 0.96.

**Figure R1.** Ordinary least squares regression of  $\sigma_{abs,660}$  from the AE33 with respect to  $\sigma_{abs,637}$  from the MAAP (New).

**Figure R2.** Ordinary least squares regression of  $\sigma_{abs,660}$  from the AE33 with respect to  $\sigma_{abs,637}$  from the MAAP (Old).

**Response Lines:** Figure S9.

**Technical corrections:**

1. 89-90: "two Magee Scientific Aethalometers"

Response: Error fixed

Response Line: 112.

1. 92-93 : please fix the intervals in the parenthesis : "(i.e. ≤ PM1, between PM1 and PM2.5, between PM2.5 and

 $PM10, \le PM10, > PM10)$ ".

Response: Fixed.

Response Lines: 115-116.

1. 106 "aethalometers"

Response: It should be aethalometer instead of aethalometers because any given time, there was only one

aethalometer operating. The AE31 was used from 2010 to 2017 and the AE33 was used from 2018 to 2022.

1. 127-128 please fix the citation format.

Response: Fixed.

Response Lines: 849-851.

1. 212 please fix the citation format.

Response: Fixed.

1. 269: "which is used in conversion of  $\sigma_{abs}$  to BC mass"

Response: Corrected.

**Response Line:** 331.

1. 277-278: The composition is not a physical characteristic

Response: Corrected to "the physicochemical characteristics of aerosols".

**Response Line:** 339.

1. 319: "contribution additional variability" this sentence is strange

Response: Corrected to "Pollen and dust events (green and red stars; Sect. 2.3) introduce additional short-term

variability in the observed optical and mass properties without altering the sign or statistical significance of the

long-term trends (Table 3)."

Response Lines: 381-384.

Fig 1 and Fig 2: Please provide the meaning of the blue shaded area on panels a and b.

**Response:** The legend has been changed.

**Response Lines:** 395-398 and 450-453.

1. 333: Please provide at least for the first notification the information on the two different values: slope and

relative trend.

**Response:** This has been done everywhere in the paper now, instead of just the first instance, so that the same

style is maintained throughout the paper.

1.475: Why "albeit"? Statistically significant is not contrasting with the beginning of the sentence.

**Response:** This section has been changed altogether.

Response Lines: 563-572.

Fig 5: It would be great to remove the decimal part of the y-axis ticks on panels c and d. Furthermore, it is a bit

difficult to see with this representation the contribution of pollen and dust events to Super PM10, as we don't

see on which months occurred these events. Maybe you can add the red and green stars also on panels c and d?

Response: The decimal part of the y-axis ticks on panels (c) and (d) have been removed. We have added the

seasonal variation of dust and pollen events in the figure.

Response Line: 680.

Response to Reviewer 2

We want to thank the reviewer for crticicall examining the manuscript. We hope that the responses below address

all your concerns.

Comment Line 225: How does autocorrelation in time series data affect the results of the Mann-Kendall test, and

how is this addressed?

Response: The classical Mann-Kendall (MK) trend test assumes serial independence. Positive autocorrelation

reduces the effective information in a series; if it is not accounted for, the MK variance is underestimated and the

test becomes too liberal (p-values spuriously small). Strong negative autocorrelation has the opposite effect. In

other words, persistence can make a weak trend appear "significant" unless the MK variance is adjusted.

We use a variance-corrected MK that replaces the nominal sample size n with an effective sample size,

$$n_{eff} = \frac{n}{1+2\sum \rho_k} (\geq 1),$$
 R2

computed from the statistically significant autocorrelation coefficients ( $\rho_k$ ) of the analysis series. Concretely, our

function effective\_sample\_size() evaluates the autocorrelation function (ACF) for lags  $k = 1...n_{lags}$  with  $n_{lags} =$

 $\min(20, \lfloor n/4 \rfloor)$  (lag 0 excluded), retains only lags with  $|\rho_k| > 1.96/\sqrt{n}$ , and then forms  $n_{eff}$ . This  $n_{eff}$  is used inside

modified\_mann\_kendall\_trend() to compute the MK variance and Z statistic, yielding autocorrelation-aware pvalues. Importantly, the trend magnitude is estimated with Theil-Sen and is unchanged by this correction; only

the significance is adjusted. We do not prewhiten, thereby avoiding potential slope bias.

- (i) Pseudo-daily variables—PM mass, MSC550, MAC520: in plot\_trend\_ps,eudo\_daily() we
- (a) mask day-level outliers using a modified Z-score, that is flag  $x_i$  if  $|0.6745(x_i \text{median})/\text{MAD}| > 3.5$ ,
- (b) estimate the slope with Theil-Sen, and
- (c) test significance with the effective sample size (ESS)—corrected Mann–Kendall test described above. Any day/interval completeness gates are enforced upstream when the pseudo-daily series are constructed; flagged pollen/dust events are annotated for context but not removed.
- (ii) Monthly-median variables— $\sigma_{sca,550}$ ,  $\sigma_{abs,520}$ ,  $SSA_{550}$ , SAE, AAE: we first form monthly medians only when a month has  $\geq$ 75%age valid hours (based on the native sampling interval), which reduces short-lag persistence; we then fit Theil–Sen and apply the same ESS-corrected MK to the monthly series.

Across both paths, the slope estimator (Theil–Sen) provides a robust trend magnitude, while the MK p-values are explicitly corrected for autocorrelation. This prevents inflated significance due to persistence and yields a conservative, reproducible assessment of trend detection appropriate for environmental time series.

**Comment Line 227:** How do seasonal fluctuations and non-linear trends influence the interpretation of long-term aerosol optical trends?

**Response:** Seasonal patterns can skew how long-term change looks. If the size or timing of the yearly cycle shifts—such as weaker winters, earlier or larger spring pollen or dust peaks, or stronger warm-season SOA with frequent spring—autumn NPF—a straight line through the raw series will absorb that movement. The apparent slope can become too big or too small, and in rare cases flip sign. These patterns also create persistence, so ignoring serial correlation can overstate confidence; our inference targets the slow component and uses statistically robust, dependence-aware trend tools (Mann, 1945; Kendall, 1975; Hamed and Rao, 1998; Yue and Wang, 2004).

Non-linear behavior brings a different challenge. Curvature across years, step changes around instrument transitions, and short pollen or dust episodes split the record into periods that do not behave the same way. One constant slope across the full series then averages dissimilar regimes and can misstate what is happening now, even when the overall fit appears good. In practice this can bias both the estimated magnitude and its assessed significance, or shift the mean without altering direction. That is why we examine shape and regime changes before summarizing the record with a single tendency. Effects matter for attribution and interpretation.

We reflect those issues in the analysis pipeline. For PM mass,  $MSC_{550}$ , and  $MAC_{520}$  we build pseudo-daily means by keeping days with  $\geq 18$  time-stamped observations, masking outliers with a modified Z-score, and averaging reliably. For the  $\sigma_{sca,550}$ ,  $\sigma_{abs,520}$ ,  $SSA_{550}$ , SAE, and the AAE, we compute monthly medians when a month has  $\geq 75\%$  valid hours, masking monthly outliers. We report the trend using the robust median slope (Theil, 1950; Sen, 1968) and test it with a statistically conservative Mann–Kendall adjusted for autocorrelation by shrinking the effective sample size using significant ACF lags up to  $n\_lags = \min(20, \lfloor n/4 \rfloor)$  (Hamed and Rao, 1998; Yue and Wang, 2004).

Comment Line 40: "which could explain higher concentrations in winter." Add "higher coarse particle

concentrations in winter."

**Response:** Now, the introduction section has been changed, so this line is redundant.

Comment Line 41-42: "while pollen and spores often originate from more local biogenic emissions that are

highly episodic and seasonal." The phrase "more local" can be improved.

**Response:** Same as above.

Comment Line 43: "Sea salt, though typically associated with marine environments, can occasionally reach

boreal forests during strong winds." Could be written as "can occasionally be transported to boreal forests during

strong wind events."

**Response:** Same as above.

Comment Line 45: Replace "over 10 μm" (line 45) with >10 μm for consistency with earlier notation (e.g., line

23: >1  $\mu$ m).

**Response:** Corrected.

Response Line: 46.

Line 45-47: "may not be fully captured by standard PM10 measurements, leaving their contributions

underrepresented...". Add the reason here "due to cut-off inlets or sampling loss".

Response: Line changed to, "Coarse-mode aerosol particle sizes span about 1 μm to > 10 μm, so many pollen

grains and fungal spores exceed the PM10 impactor cut-off and are underrepresented in PM10 measurements

(Després et al., 2012; Yli-Panula et al., 2009)."

Line 49:"contribute to atmospheric heterogeneity". Specify "spatial and temporal heterogeneity" for more

scientific clarity.

**Response:** Corrected as advised.

Response Line: 65.

Comment Line 51: the reference added here is in non-chronological citation order. Also, Brasseur et al., 2024 is

cited in a 2025-dated document. While plausible (if published in early 2024), ensure this reference exists. If not,

update to the correct publication year.

**Response:** I have used a citation manager tool that created citations as per the author's last name. That is why

Brasseur appears first. Also, the Brasseur et al. was published in 2024 and the link to the correct paper is included

in the reference list.

Response Line: 68.

Comment Line 57: "potentially leaving a significant fraction of aerosol mass unquantified". Replace with

"potentially resulting in a significant underestimation of aerosol mass."

Response: Done.

Comment Lines 64-68: These line are slightly redundant with earlier statements and too long

**Response:** This paragraph has been rewritten.

Response Lines: 80-91.

Comment Line 86-87: "Thus, SMEAR II represents the typical conditions that may be found in a boreal forest."

Should be replaced by "Therefore, SMEAR II reflects typical boreal forest conditions."

**Response:** Changed as advised.

Response Line: 108.

Comment Line 88: "It is a part of the European Aerosols, Clouds, and Trace Gases Research Infrastructure or

ACTRIS..." should be replaced by "The station is part of ACTRIS (Aerosols, Clouds, and Trace Gases Research

Infrastructure)..."

Response: Done.

Response Line: 111.

Comment Line 60: "such as pollen, may escape standard measurements, whereas smaller coarse-mode particles,

such as fungal spores and dust, are more likely to be captured. Replace with "Smaller coarse-mode particles like

fungal spores and dust are more likely to be captured, whereas larger ones, such as pollen, may escape detection."

Response: Corrected.

Response Line: 76-77.

**Comment Line 77:** Define "super-PM10" earlier for clarity:37-48: Provide references for these statements.

Response: Done.

**Response Lines:** 81-85.

Comment Line 88-93: 4 instruments have been used primarily, but only 3 have been covered here.

Response: Changed.

Response Lines: 108-117.

Comment Line 97: 'a' Dp >10 µm

**Response:** Done.

Response Line: 120.

Comment Line 104: is aimed 'to' be kept

Response: Corrected.

Comment Line 120: Are there any recent publications post Liousse et al., 1993 which cover this?

Response: Yes and they have been added.

Response Lines: 140-150.

Comment Line 149: Use 1 min-1 for consistency.

Response: Changed everywhere.

Comment Line 237-238: This is repetition of 232-233.

Response: Changed.

Response Lines: 299-300.

Comment Line 289: Write PM1–10 with subscript.

**Response:** Changed everywhere.

Comment Line 341: SOA already defined in line 33, so expansion is not needed here.

Response: Corrected.

Response Line: 409.

Comment Line 342: Instead of 'biogenic VOC', use BVOC as already defined.

Response: Done.

Response Line: 410.

Comment Line 460: OA has not been defined previously.

**Response:** This text has been removed.

Comment Line 645: SOA already defined in line 33, so expansion is not needed here.

**Response:** This text has now been removed altogether.

Comment Line 709: typo after reference link

Response: Changed.

Response Lines: 840-844.

**Comment Line 760:** Provided link is not working.

**Response:** Corrected.

Response Line: 891.

Comment Line 927: Link to reference has not been provided. Pls correct all the incorrect references.

Response: The link to the reference has been provided. All the incoreect references have been corrected.

**Response Line:** 1043.

**Comment:** Use subscript formatting for PM size ranges (e.g.,  $PM_{1-10}$  instead of PM1-10).

**Response:** Changed everywhere.